# The type IV pilus chemoreceptor PilJ controls chemotaxis of one bacterial species towards another

Kaitlin D. Yarrington[1], Tyler N. Shendruk[2], Dominique H. Limoli[1,3]*

**1** Department of Microbiology and Immunology, Carver College of Medicine, University of Iowa, Iowa City, Iowa, United States of America, **2** School of Physics and Astronomy, The University of Edinburgh, Edinburgh, United Kingdom, **3** Department of Biology, Indiana University, Bloomington, Indiana, United States of America

* dlimoli@iu.edu

**Data Availability Statement:** All relevant data are within the paper and its Supporting Information files.

## Abstract

Bacteria live in social communities, where the ability to sense and respond to interspecies and environmental signals is critical for survival. We previously showed the pathogen *Pseudomonas aeruginosa* detects secreted peptides from bacterial competitors and navigates through interspecies signal gradients using pilus-based motility. Yet, it was unknown whether *P. aeruginosa* utilizes a designated chemosensory system for this behavior. Here, we performed a systematic genetic analysis of a putative pilus chemosensory system, followed by high-speed live-imaging and single-cell tracking, to reveal behaviors of mutants that retain motility but are blind to interspecies signals. The enzymes predicted to methylate (PilK) and demethylate (ChpB) the putative pilus chemoreceptor, PilJ, are necessary for cells to control the direction of migration. While these findings implicate PilJ as a bona fide chemoreceptor, such function had yet to be experimentally defined, as full-length PilJ is essential for motility. Thus, we constructed systematic genetic modifications of PilJ and found that without the predicted ligand binding domains or predicted methylation sites, cells lose the ability to detect competitor gradients, despite retaining pilus-mediated motility. Chemotaxis trajectory analysis revealed that increased probability and size of *P. aeruginosa* pilus-mediated steps towards *S. aureus* peptides, versus steps away, determines motility bias in wild type cells. However, PilJ mutants blind to interspecies signals take less frequent steps towards *S. aureus* or steps of equal size towards and away. Collectively, this work uncovers the chemosensory nature of PilJ, provides insight into how cell movements are biased during pilus-based chemotaxis, and identifies chemotactic interactions necessary for bacterial survival in polymicrobial communities, revealing putative pathways where therapeutic intervention might disrupt bacterial communication.

## Introduction

Microbes often exist in complex, dynamic environments and have evolved sophisticated systems to perceive and respond to the outside world. Because bacteria commonly reside in

**Funding:** This work was supported by funding from the Cystic Fibrosis Foundation (https://www.cff.org/): CFF Postdoc-to-Faculty Transition Award LIMOLI18F5 (DHL), CFF RDP Junior Faculty Recruitment Award LIMOLI19R3 (DHL), and CFF Student Traineeship Award YARRIN21H0 (KDY), funding from the National Institutes of Health (https://www.nih.gov/):R35GM142760 (DHL), and funding from the European Council (https://erc.europa.eu/homepage): European Union's Horizon 2020 research and innovation programme Grant Agreement No. 851196 (TNS). The funders had no role in study design, data collection and analysis, decision to publish, or preparation of the manuscript.

**Competing interests:** The authors have declared that no competing interests exist.

**Abbreviations:** cAMP, cyclic adenosine monophosphate; CCW, counterclockwise; CW, clockwise; LB, lysogeny broth; LBD, ligand binding domain; MCP, methyl-accepting chemotaxis protein; MSD, mean-squared displacement; PBS, phosphate buffered saline; PDF, probability density function; PSM, phenol soluble modulin; TFP, type IV pilus; TSB, tryptic soy broth.

multispecies communities, they experience gradients of nutrients, metabolites, and secreted factors generated by neighboring cells. Gradients are particularly steep in surface-attached biofilm communities and ecological theory predicts bacteria must sense and respond to competitor and cooperator signals to thrive in such complex environments [1–2].

In line with this hypothesis, we recently reported that *Pseudomonas aeruginosa* is attracted to gradients of secreted factors from other microbial species [3]. Pseudomonads are opportunistic bacteria found in polymicrobial communities in soil, wounds, and chronic lung infections, such as those in people with cystic fibrosis [4–5]. *P. aeruginosa* is frequently coisolated with *Staphylococcus aureus* from cystic fibrosis respiratory samples, and coinfections can persist for decades [4,6–8]. Coinfection is also associated with pulmonary decline; thus, understanding ecological competition between these organisms may provide insight into patient outcomes [9–11].

Accordingly, in vitro studies have documented interspecies interactions between *P. aeruginosa* and *S. aureus* leading to reciprocal enhancement of antibiotic tolerance, production of virulence factors, and the ability to alter host immune cell responses, further supporting clinical observations [4,12–14]. Additionally, these data suggest each species may sense a secreted signal from the other, which instigates a competitive or cooperative response through alteration of their virulence arsenals, a model supported by differential regulation of specific *P. aeruginosa* virulence pathways in response to *S. aureus* exoproducts [15,16]. Remarkably, *P. aeruginosa* and *S. aureus* have been shown to form mixed microcolonies when cocultured on bronchial epithelial cells [13]. One possible explanation for the formation of mixed communities is that *P. aeruginosa* and *S. aureus* may be initially attracted to one another through detection of secreted interspecies signals. Such attraction has the potential to facilitate formation of blended microcolonies or microbial competition, depending on the environmental conditions.

Supporting this model, *P. aeruginosa* senses secreted *Staphylococcal* peptide toxins referred to as phenol soluble modulins (PSMs) and responds with directed motility towards the increasing PSM concentration gradient, mediated by the type IV pilus (TFP) [3]. With the identification of a putative interspecies chemoattractant for pilus-based motility, we hypothesize that *P. aeruginosa* uses a chemosensory pathway to move towards *S. aureus*. TFP-mediated motility, or twitching motility, occurs through the grappling hook activity of the pilus, which undergoes episodes of extension, substrate attachment, and retraction, which pulls the cell body along the surface [17]. The direction of twitching motility is thought to be controlled by preferential extension of pili at the pole facing the direction of movement, referred to as the leading pole. Cells are predicted to change direction by extending pili from the opposite pole, reversing the direction of cellular movement along the long axis of the cell body and swapping leading poles [18]. However, whether modulation of reversal frequency is necessary and sufficient to bias the movement of twitching cells towards a chemoattractant, similar to the run-and-tumble or run-reverse-turn strategies used in flagella-mediated chemotaxis, remains unknown [19]. While planktonic swimming cells smoothly sample gentle chemoattractant gradients by typically traveling a full body length or more between tumbling events, surface-associated twitching motility is hundreds to thousands of times slower and steep, varying gradients characterize the chemotactic landscape [20,21]. Therefore, twitching cells experience less certain chemotactic signals [2,22,23]. Thus, we predict that additional parameters need to be considered to fully understand how a twitching community biases directional movement. Indeed, even simple models of twitching motility are known to produce complex dynamics [24].

While TFP-mediated chemotaxis has not been thoroughly dissected in *P. aeruginosa*, prior work has described chemotactic roles for some proteins of the putative pilus chemosensory system, Pil-Chp [2,18]. Namely, the predicted CheY-like response regulators, PilG and PilH,

are thought to control reversals and increase levels of the intracellular second messenger cyclic adenosine monophosphate (cAMP) through PilG activation of the adenylate cyclase CyaB [2,18,25,26]. cAMP controls a large arsenal of virulence factors targeting both eukaryotic and prokaryotic cells, as well as multiple modes of motility through activation of the virulence response transcription regulator, Vfr [27]. However, whether cAMP is also necessary to transduce the detection of interspecies attractants to modulate directional motility has not been investigated.

In addition, Pil-Chp includes homologous proteins to a majority of the CheI flagella chemotaxis system in *P. aeruginosa* [28,29]. However, unlike the 24 CheI-associated chemoreceptors, referred to as methyl-accepting chemotaxis proteins (MCPs), Pil-Chp only has one MCP, called PilJ. This putative pilus chemoreceptor differs from the flagella systems, in that PilJ is uniquely essential for twitching motility and possesses low protein sequence similarity in both the ligand binding domain (LBD) and cytoplasmic signaling domains with other MCPs [28,30]. The chemoreceptor for this system and the chemotactic role of the rest of the pathway has yet to be fully interrogated.

To determine the contribution of the remaining proteins encoded within the Pil-Chp pathway, we systematically deleted these genes and identified those mutants that retain twitching yet are unable to bias movement up a gradient of *S. aureus* secreted factors. From this analysis, 6 genes fit these criteria: *pilK*, *chpB*, *chpC*, *pilH*, *cyaB*, and *cpdA*. ChpC is a CheW-like linker protein and PilH is a response regulator, while CyaB and CpdA are enzymes that synthesize and degrade cAMP, respectively. PilK and ChpB are proteins that control chemoreceptor adaptation through methylation of the Pil-Chp MCP PilJ, implicating a role for PilJ as a chemoreceptor for interspecies signals. Accordingly, PilJ mutations in the key regions that define an MCP, including the LBD for sensing interspecies signals and methylation sites for chemoreceptor adaptation, revealed that PilJ is necessary for *P. aeruginosa* to perceive and bias movements towards *S. aureus*. Tracking analysis of each chemoreceptor-associated mutant uncovered differential step size and frequency of steps when moving towards *S. aureus* as a major determinant for properly biasing pilus-mediated motility. Quantification of cAMP in single cells also reveals that cAMP levels rise in *P. aeruginosa* during chemotaxis towards *S. aureus*; yet, cAMP levels are not strictly due to enhanced twitching motility. Collectively, these results define a novel chemosensory role for PilJ to sense *S. aureus* PSMs via its LBDs and signal downstream through the Pil-Chp system to respond to these interspecies signals.

## Results: The Pil-Chp system controls *P. aeruginosa* attraction to *S. aureus* peptides

The directional nature of *P. aeruginosa* movement up a gradient of *S. aureus* secreted peptides suggests a role for a chemosensory network. We hypothesize that Pil-Chp controls a chemotaxis-like TFP-mediated response by *P. aeruginosa* towards *S. aureus*. To test this hypothesis, we systematically deleted genes in several components of the Pil-Chp pathway to identify mutants that retain twitching motility but show diminished directional response to *S. aureus*, using a macroscopic directional motility assay (Fig 1A and 1B) [3,31]. Prior studies revealed PSMs as the primary component of *S. aureus* supernatant necessary to attract *P. aeruginosa*; therefore, in initial experiments, cell-free *S. aureus* supernatant containing PSMs was spotted on top of an agar plate and allowed to diffuse for 24 hours to form a gradient and compared to growth medium alone [3,32]. In later experiments, synthetic PSMs will also be used to confirm the necessity of Pil-Chp to respond to *S. aureus* peptides. *P. aeruginosa* was then spotted at a distance from the gradient and allowed to respond to each gradient for 36 hours before imaging the plates and quantifying the directional motility ratio (Fig 1C). Four of the 16 pilus

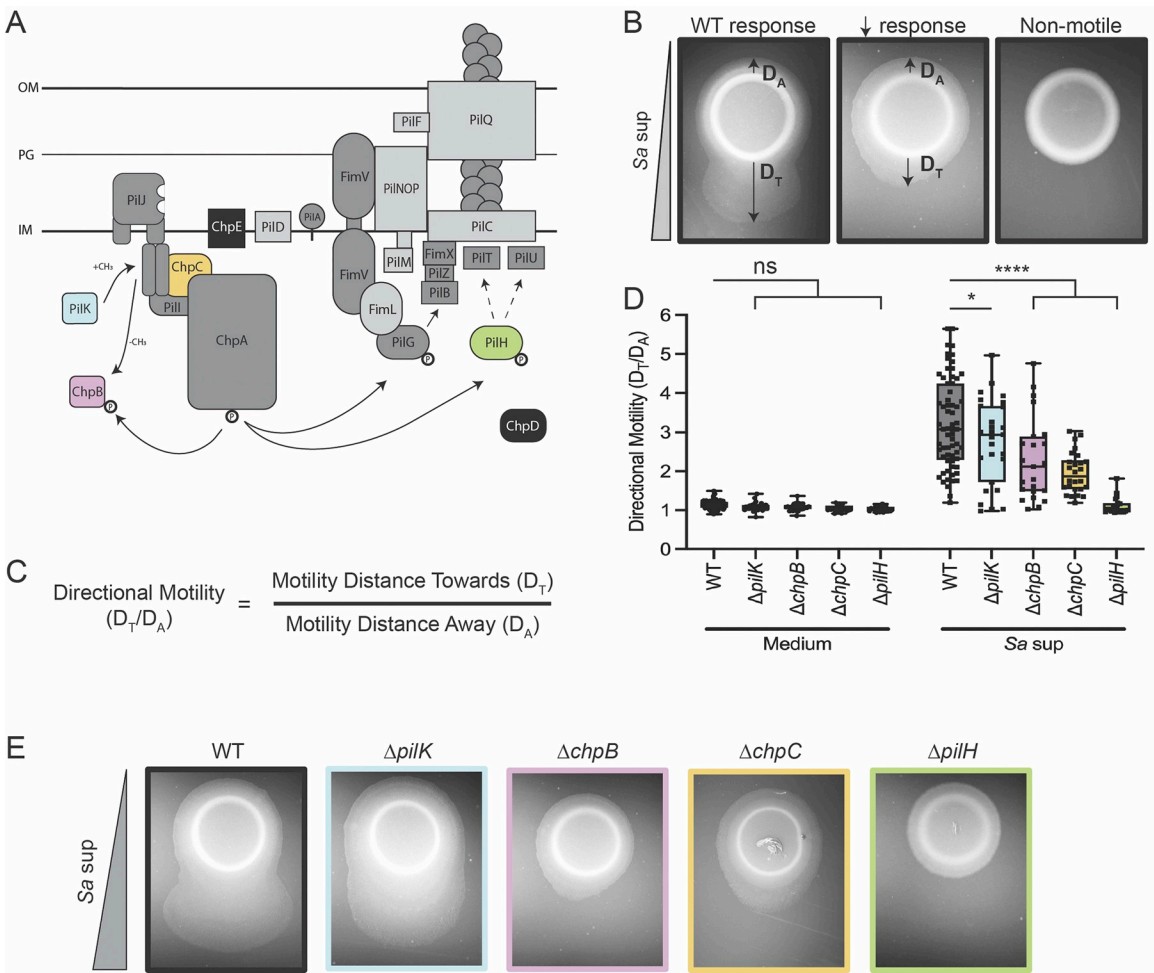

**Fig 1. The Pil-Chp system controls *P. aeruginosa* attraction to *S. aureus* peptides.** (**A**) Schematic of the putative TFP Pil-Chp chemosensory pathway. When the colored proteins are deleted, cells retain TFP motility but show reduced pilus-mediated response to the gradient of *S. aureus* secreted factors. Mutants for the proteins highlighted in black retain TFP motility and wild type levels of response up a gradient of *S. aureus* secreted factors. When the proteins highlighted in dark gray are deleted, the cells are nonmotile. Proteins highlighted in light gray were not tested here but have been previously reported to lack TFP motility. Arrows show the flow of the signaling cascade through Pil-Chp and do not necessarily indicate direct interactions between proteins. Representative images for each response type are shown in (**B**). (**C**) Response measured by calculating the ratio of directional motility up the gradient of *S. aureus* secreted factors. The equation for calculating directional motility is shown. (**D**) Directional motility of *pil-chp* mutant candidates that retain TFP motility but show reduced response to *S. aureus* supernatant (*Sa* sup) and representative directional motility images for the wild type and each mutant shown in (**E**). Directional motility for at least 3 biological replicates, each containing a minimum of 4 technical replicates are shown and statistical significance determined with a two-way ANOVA followed by Dunnett's multiple comparisons test. **** indicates $p < 0.0001$; * indicates $p < 0.05$; ns indicates no statistically significant difference in directional motility compared to wild type *P. aeruginosa*. The underlying data can be found in S1 Data. IM, inner membrane; OM, outer membrane; PG, peptidoglycan; TFP, type IV pilus; WT, wild type.

mutants retained twitching motility but reduced directional motility up a gradient of *S. aureus* secreted signals (Fig 1D and 1E). The remaining mutants either phenocopy wild type or are nonmotile (Fig 1A and 1B). Mutants with reduced ability to respond include genes that encode for the predicted methyltransferase PilK and methylesterase ChpB. Additionally, the mutants for ChpC, which is a CheW-like linker protein that connects the chemoreceptor and kinase, and PilH, which is a response regulator that is predicted to regulate pilus retraction, lacked complete response to *S. aureus* [33–35]. These 4 mutants indicate that *P. aeruginosa* may use the Pil-Chp system to modulate pilus-mediated chemotaxis towards *S. aureus*. Yet, how these

chemotaxis-deficient mutants control *P. aeruginosa* attraction towards *S. aureus* is not precisely defined by macroscopic directional motility assays.

## Results: Methyl modification proteins for chemotaxis adaptation are necessary for full directional TFP-mediated motility towards *S. aureus*

Since both predicted sensory adaptation proteins, PilK and ChpB, were necessary for full response to *S. aureus* at the community level, we next investigated the behavior of each mutant in more detail to uncover how each is unable to correctly bias the direction of movement. PilK is predicted to methylate, while ChpB is predicted to demethylate PilJ (Figs 1A and 2A). For flagella-associated MCPs, the addition and removal of methyl groups to a chemoreceptor facilitates intracellular signal transmission to the downstream kinase to shift activity towards more "ON" or "OFF" states, while also inducing conformational changes altering signal transmission sensitivity [36]. By analogy, higher levels of PilJ methylation facilitated by PilK are expected to alter chemoreceptor sensitivity to input signals and elevate ChpA kinase activity. Meanwhile, hydrolysis of methylated sites by ChpB is expected to reduce chemoreceptor sensitivity and subsequent kinase activity (Fig 2A). By this model, to adapt to chemoeffector gradients, cells require the ability to dynamically modulate the methylation state of the chemoreceptor, and we predict cells lacking either PilK or ChpB would exhibit aberrant chemosensory adaptation (Fig 2A). Accordingly, in the macroscopic chemotaxis assay, both the *pilK* and *chpB* mutants displayed altered directional movement towards *S. aureus* secreted factors, which was restored upon complementation (Figs 1 and S1). Deletion of *chpB* produced a dramatic defect in directional motility, suggesting methyl hydrolysis is critical for cells to modulate TFP chemotaxis (Fig 1D and 1E).

To uncover how Δ*chpB* fails to move fully towards *S. aureus* signals, we evaluated single-cell TFP-mediated motility behaviors of Δ*chpB* in the absence and presence of *S. aureus*. Compared to wild type *P. aeruginosa*, in both monoculture and coculture, the Δ*chpB* mutant exhibits increased motility away from the growing microcolony, with groups of motile cells migrating outwards in all directions (S1–S4 Videos). In coculture with *S. aureus*, some cells migrate towards *S. aureus*; however, similar numbers of cells move in the opposite directions, suggesting Δ*chpB* cells are unable to bias movements towards the interspecies signals, unlike wild type, which shows stronger apparent bias with more cells moving towards *S. aureus* (S4 Video).

To quantify these behaviors, we first determined the direction of motion for each *P. aeruginosa* cell relative to its starting position. These principal angles of motility for all cell trajectories were plotted to generate rose graphs (Fig 2B). Vector wedges on the rose graph grow longer away from the center and darker with an increasing number of cells sharing the same principal angle of motion. After plotting trajectories for all motile cells from a microcolony, the dominant direction moved by the group was designated as the parallel positive direction (∥ +; Fig 2B, dotted lines; Methods and materials). Based on the angle assigned as parallel positive, 3 other trajectory directions were also defined: the parallel negative, 180° from the parallel positive, and the perpendicular positive and perpendicular negative, which, respectively, are 90° to the right and left of the parallel positive direction (Fig 2B, left schematic). Once all trajectories were plotted and the dominant (∥+) direction identified, each *P. aeruginosa* strain could be assessed to determine if this direction was towards *S. aureus* or in another direction. On each rose graph, a *S. aureus* cluster cartoon represents the angle that the *S. aureus* microcolony was positioned relative to the center of the *P. aeruginosa* microcolony (Fig 2B). Analysis of the completed rose graphs reveal that wild type *P. aeruginosa* cells have a parallel positive direction

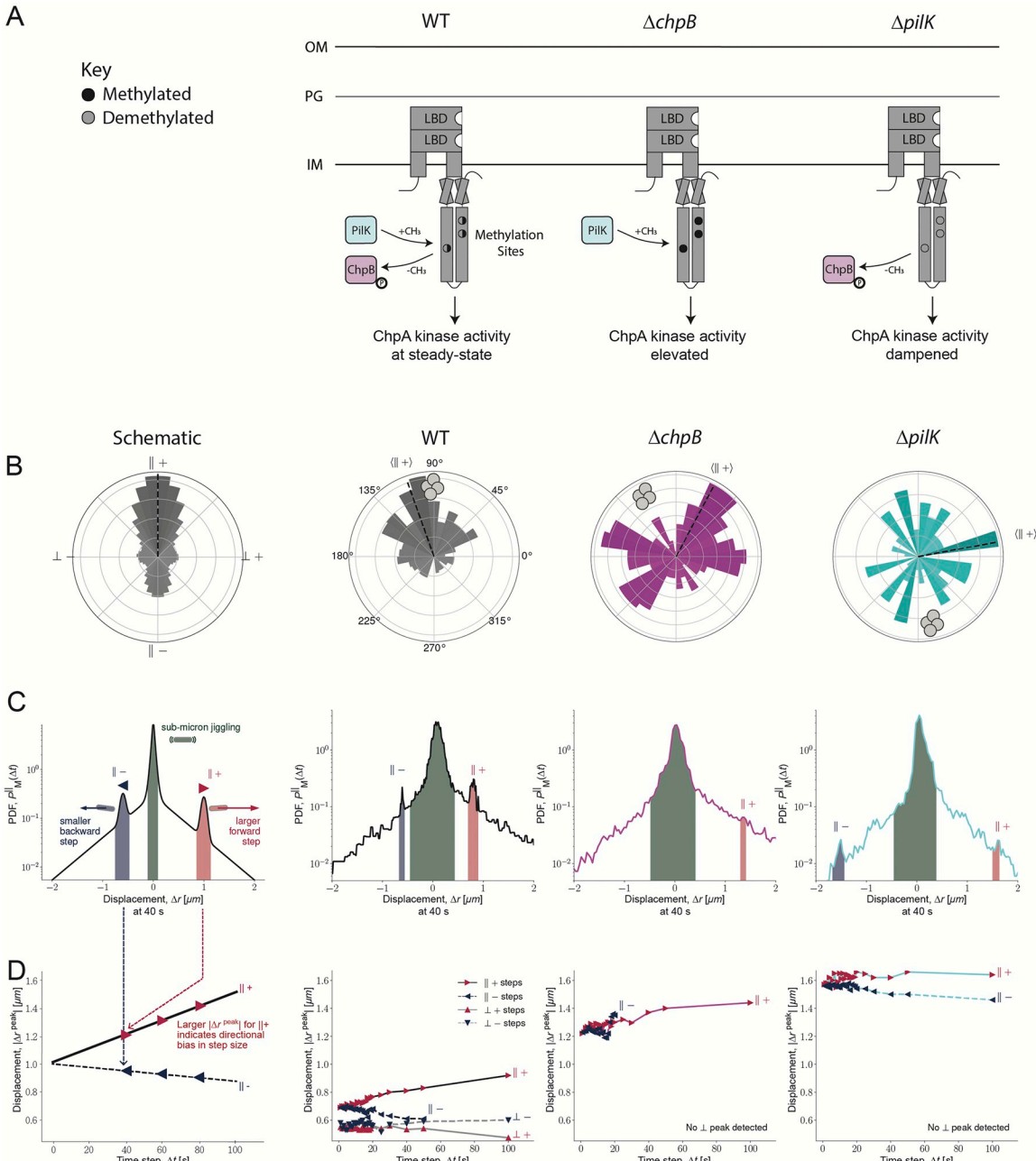

**Fig 2. Methyl modification proteins for chemotaxis adaptation are necessary for full directional TFP-mediated motility towards *S. aureus*.** (**A**) Schematic of *P. aeruginosa* MCP protein PilJ and methyl modification proteins PilK and ChpB. The predicted methylation sites with cytoplasmic domain of PilJ are represented by the black and gray circles. In wild type, PilJ is predicted to undergo methylation and demethylation; thus, ChpA kinase activity shifts between more ON or OFF states (left). In the absence of *chpB*, PilJ is expected to have high methylation (filled, black circles), and, therefore, ChpA kinase activity shifted ON (middle). In the absence of *pilK*, PilJ is expected to have low methylation (empty, gray circles), and, therefore, ChpA kinase activity shifted OFF (right). For each analysis in B-D, an annotated schematic is shown in the leftmost column, followed by (left-to-right) *P. aeruginosa* WT, Δ*chpB*, and Δ*pilK* in coculture with *S. aureus*. (**B**) Rose graphs of the principal angle of motility for each cell trajectory relative to starting position. Position of *S. aureus* relative to the center of the *P. aeruginosa* microcolony is represented by the gray cocci on the perimeter. Trajectory angles are shown by colored vectors with the average angle of all trajectories, which determines the parallel positive direction (||+), represented by the dotted black line. Longer, darker vectors indicate more cells for the given principal angle. One rose graph representative of at least 3 replicates is shown. (**C**) PDFs of step sizes (Δ*r*) parallel to the direction of motion (||) of moving (M) wild type, Δ*chpB*, and Δ*pilK* cells. Each PDF is for a time step (lag time) of Δ*t* = 40 seconds and exhibits a peak of highly probable sub-micron jiggling (green region). The blue and red regions highlight the nonzero sharp-shoulder peak step size in the backward (||−) and forward (||+) directions. (**D**) Peak displacements (μm) over various time steps for forward and backwards displacements in both parallel and perpendicular directions. The

value for each plotted step size (red and blue triangles) at a given time step corresponds with the most probable displacement value in the respective relative distribution plots (graphs in C and S3 Fig), as shown in the schematic on the left. While all strains exhibit parallel and perpendicular motion, not all strains have sharp-shoulder peak values in each direction. The underlying data can be found in S2–S4 Data. IM, inner membrane; LBD, ligand binding domain; MCP, methyl-accepting chemotaxis protein; OM, outer membrane; PDF, probability density function; PG, peptidoglycan; TFP, type IV pilus; WT, wild type.

angled closely to the position of *S. aureus* (Fig 2B, middle left, gray). This demonstrates that wild type *P. aeruginosa* cells bias their collective direction of movement towards *S. aureus*, quantifying their ability to chemotax towards interspecies signals. In contrast, *P. aeruginosa* Δ*chpB* does not exhibit a directional bias towards *S. aureus*; instead, Δ*chpB* cells are just as likely to move away, or transverse, from *S. aureus*, as they are to move toward, therefore not displaying directed chemotactic behavior (Fig 2B, middle right, pink).

Despite loss of controlled motility biased towards interspecies signals, the enhanced single-cell motility of *chpB* supports the prediction that loss of ChpB may increase pilus activity. However, it is unclear whether the loss of directional movements by the *chpB* mutant is strictly due to hypermotility or the result of chemosensing defects. To help uncover what aspects of *P. aeruginosa* movement contribute to biased motion towards *S. aureus*, dynamics of the trajectories were initially quantified by the mean-squared displacement (MSD), a measure of the average distance traveled over a given time step $\Delta t$ (S2 Fig). As the cells move, MSD increases over time. Importantly, how the MSD increases exposes whether cells move in a direct manner or in a random walk. If performing random walks, the MSD is proportional to the time step ($\Delta t$), which is written as MSD $\sim \Delta t^1$ with exponent $\alpha = 1$. However, when cells propel themselves with constant directed motion, the MSD is now written as MSD $\sim \Delta t^2$ and the exponent increases to $\alpha = 2$ (Methods and materials; Eq 4). If *P. aeruginosa* cells bias movement towards *S. aureus*, one expects $\alpha \geq 1$, greater than a random walk but still much smaller than 2, that is, not strictly propulsive (not a beeline towards *S. aureus*). Further, MSD measurements commonly assume that motion in each direction is equivalent; therefore, the trajectories were again split into parallel and perpendicular directional components relative to each trajectory's principal direction (||+) for analysis (Methods and materials; Eq 3).

While wild type cells travel comparable distances in the parallel and perpendicular directions (S2 Fig), the dynamics in each direction are different. This can be seen because the parallel and perpendicular displacements gradually separate over longer time durations, illustrated by their dissimilar exponents, which represent different dynamics in the 2 directions (S2 Fig, left). In the perpendicular direction, the cells perform an unbiased random walk, which is quantified by fitting the exponent (Methods and materials; Eq 4) and finding $\alpha_\perp = 1.1$, which is close to the random-walk value of 1. However, in the parallel direction, the exponent $\alpha_\parallel = 1.3$, and this larger value suggests that cells tend to move with more self-directed, propulsive transport in the parallel direction.

In comparison to wild type, cells lacking ChpB exhibit markedly different MSD profiles. Δ*chpB* cells have significantly larger displacements in the parallel direction than perpendicular (S2 Fig, middle). While this is true of all time steps, it is well illustrated by the values at $\Delta t = 20$ seconds, when the perpendicular MSD is only $0.12 \pm 0.03$ $\mu m^2$ but the parallel MSD is $0.27 \pm 0.10$ $\mu m^2$, more than twice as large. However, this increased microscopic motion does not translate into directional motility, since the exponent is near unity in both the parallel ($\alpha_\parallel = 1.1$) and perpendicular ($\alpha_\perp = 1.0$) directions, which suggests random walk dynamics. The combination of increased MSD, but loss of self-directed, propulsive transport, as quantified by $\alpha$ and visualized on the rose graph, compared to wild type is consistent with reduced regulation of directed chemotactic motility (Figs 2B and S2).

Given the predicted chemotactic role for Pil-Chp, one possible explanation for reduced bias towards *S. aureus* is that Δ*chpB* cells have increased rates of cellular reversals. Reversing, or changing the direction of type IV pilus-mediated movements, is thought to require switching of the leading and lagging poles [37,38]. These cellular reversals also enable bacteria to bias the direction of type IV pilus-mediated movement up an increasing concentration gradient of chemoattractant [2]. However, under the imaging conditions here, reversals were rarely observed in any *P. aeruginosa* strain, and single-cell tracking analyses did not show significantly different reversal dynamics between Δ*chpB* and wildtype. Therefore, if cellular reversals do not explain how cells can perform TFP-mediated chemotaxis towards interspecies signals, then how are cells able to bias their movements up a gradient?

One possibility is that twitching cells undergo increased pilus activity upon sensing higher concentrations of chemoattractant, leading to more pilus extension and retraction events or stronger pilus retraction as the cell body is pulled towards the increasing chemoattractant gradient. In conjunction, this would lead to enhanced pilus dynamics as long as cells move towards increased chemoattractant concentrations, but slower pilus dynamics as the surrounding concentration of signal decreases or remains constant. If chemotactic cells experience increased pilus activity, we hypothesize there would be distinct displacement events observed in the trajectories as *P. aeruginosa* moves towards rising concentrations of interspecies signals. To test this hypothesis and to understand how directional motility arises from the single-cell dynamics, we considered the displacement distribution function, which gives the probability that a cell moves a given displacement over time (Figs 2C and S3). The probability density functions of displacements made by cells of any *P. aeruginosa* strain are dominated by a narrow peak centered on zero with a width that is ≤0.1 μm (Figs 2C and S3, tallest peaks). That is, these cells undergo submicron displacements that appear as small "jiggling" movements [18], characterized as random, undirected motions that displace cells much less than a cell body width from their starting position (S3 Fig). We verify that this submicron jiggling is not simply an imaging artifact by measuring the diffusion of a nonmotile particle and see that, while imaging introduces narrow Gaussian noise (S4A Fig), the submicron jiggling peak of *P. aeruginosa* cells is wider and exponentially distributed (S4B Fig). In addition to frequent jiggling, there is an ever-present probability of rare-but-large steps, which are hundreds of times less frequent than jiggling steps, yet may be sufficient to enable directional motion, which we explore below (S4B Fig). When all cells are collectively evaluated, neither jiggling nor rare-but-large steps show substantial differences parallel or perpendicular to the principal direction of motion for wild type *P. aeruginosa*. However, there is a large portion of cells that do not move. Because nonmoving trajectories may hide the dynamics of actively moving cells, we divided cell trajectories into subpopulations of "movers" and "resters" (see Methods and materials; S3 Fig). While rester-designated cells exhibit displacement distributions with only jiggling motion, wild type movers possess step sizes in the direction parallel to the principal angle, represented by a pair of nonzero sharp-shoulder peaks (Fig 2C, middle left, blue and red peak regions, labeled ||− and ||+, respectively). These peaks represent a well-defined displacement step size of 0.69 ± 0.01 μm at the shortest time durations in the parallel directions, which is comparable to the previously reported distance a cell body moves per pilus retraction [39]. The position of the displacement peak ($|\Delta r^{peak}|$) at each time step ($\Delta t$) can be plotted together to assess changes in step sizes in all directions and across various durations of time to compare these dynamics between strains (Fig 2D, left schematic). For wild type cells, the position of the forward/positive parallel direction (||+) sharp peak grows to step sizes that approach nearly 1 μm at longer durations but shrink and eventually disappear altogether in the backward/negative parallel direction (||−; Fig 2D, middle left, and S3 Fig). This directional bias of wild type cells moving in the direction of *S. aureus* reveals how the forward (||+) and backward (||−)

peak steps become asymmetrical with increasing time durations: The forward peak shifts towards slightly larger step sizes over longer times, and the backward peak shifts to slightly smaller step size values and eventually vanishes at the largest time durations (Fig 2D, middle left, and S3 Fig, top). Meanwhile, *P. aeruginosa* wild type cells take smaller sized steps in the perpendicular directions, without developing a bias, across any duration of time (S3 Fig and Fig 2D, middle left). The probability distributions at different times reveal the microscopic basis of the directed motility of wild type *P. aeruginosa* towards *S. aureus* and the difference between these forward and backwards peak step sizes across longer times explains the more propulsive motion.

Comparable quantification of the step sizes taken by Δ*chpB* cells does not show a distinguishable difference in forward and backward displacement step sizes (Fig 2C, middle right, short blue and red peak regions, and Fig 2D). Whereas wild type cells have clear sharp-shoulder peaks (Fig 2C, middle right, tall blue and red peaks) that vary between 0.5 and 1.0 μm in size (Fig 2D, middle right), the Δ*chpB* mutant does not have well-defined shoulder peaks (Fig 2C, middle right, short blue and red peaks, and S3 Fig, middle row). Instead, at the shortest durations, Δ*chpB* exhibits a slightly broadened step size distribution, as shown by the nonzero displacement tails that extend farther out in both the parallel (∥+) and antiparallel (∥−) directions (S3 Fig). The nonzero sharp-shoulder peaks are less frequent and broadened compared to wild type (S3 Fig) but represent larger step sizes (Fig 2D, middle right). While still rare events, the displacement distribution shows Δ*chpB* has a higher probability of taking the largest steps in the positive direction, with a step size greater than 1 μm, compared to wild type (Fig 2C, middle). Closer examination of the peak step sizes for Δ*chpB* reveal that the most frequent steps at any duration are larger, with the smallest peak Δ*chpB* step size value exceeding the largest peak step size observed for wild type (Fig 2D, middle); however, the probability of taking a step is much lower in any direction and entirely absent in the perpendicular direction (Fig 2C and 2D, middle right). These observations can be seen in the live-imaging of Δ*chpB* where cells exhibit sub-micron jiggling, occasionally take a large step, then return to sub-micron jiggling, before another large step is taken (S4 Video). Further, the large Δ*chpB* steps occur in multiple directions, not exclusively towards *S. aureus* (S4 Video). Collectively, these analyses suggest that ChpB is necessary for *P. aeruginosa* to coordinate step sizes and properly bias directed movements towards *S. aureus*.

Given that reduced pilus-mediated chemotaxis of Δ*chpB* can be explained by a lack of frequent, biased steps towards *S. aureus*, we next live-imaged Δ*pilK* and tracked cells to evaluate whether Δ*pilK* exhibits similar reduction in frequency, size, and bias, thus explaining the modest macroscopic chemotaxis deficiency. Interestingly, single-cell imaging of Δ*pilK* in monoculture and coculture with *S. aureus* revealed 2 contrasting phenotypes. While 50% of Δ*pilK* microcolonies imaged show hypermotile single-cells or small packs of cells moving outwards in all directions from the growing microcolony, regardless of where *S. aureus* is in coculture, the other 50% of imaged Δ*pilK* microcolonies do not show any motility (S5 Fig and S5–S8 Videos). We hypothesized that this bimodal motility phenotype could either be the result of stochastic single-cell heterogeneity or be a consequence of increased susceptibility to small changes in environmental conditions. To differentiate these potential explanations, we imaged multiple microcolonies originating from a single cell in multiple culture dishes. We predicted that if the bimodal phenotype results from heterogeneity among microcolony founder cells, we would observe variation among microcolonies in the same dish. Instead, we observed a similar behavior among microcolonies in the same dish, but heterogeneity in behaviors dish-to-dish. These observations suggest that the motility of cells lacking PilK is more susceptible to subtle changes in environmental conditions.

Focusing on the motile $\Delta pilK$ cells, we observed cells disperse from the microcolony later than $\Delta chpB$ but earlier than wild type (S6 Video). Compared to $\Delta chpB$ cells, which tend to move as clusters of cells in tendril-like patterns away from the microcolony, $\Delta pilK$ cells more frequently move in smaller groups or strictly as single cells and spread outwards radially (S6 and S8 Videos). The more uniform departure from the microcolony is exemplified by the rose graph trajectories, where vector wedges are more equal in length and more evenly distributed among all angles compared to wild type or $\Delta chpB$ (Fig 2B, right). Additionally, while a principal angle of motion is computationally designated based on the trajectories, this direction is clearly not pointed towards the *S. aureus* microcolony nor is it the angle that a majority of cells predominantly move, in contrast to wild type and, to lesser extent, $\Delta chpB$ (Fig 2B, right). Similar to $\Delta chpB$, the parallel anomalous exponent of the $\Delta pilK$ trajectories, $\alpha_\| = 1.1$, suggests $\Delta pilK$ pilus-mediated movements generate a diffusive random walk rather than propulsive motions, even though the MSD resembles values more akin to wild type (S2 Fig).

Single-cell imaging of $\Delta pilK$ features a similar pattern of motility as $\Delta chpB$, consisting of sub-micron jiggling with intermittent large steps (S6 and S8 Videos). Comparison of the step size probabilities for these cells show a similarly broadened parallel ($\|+$) and antiparallel ($\|-$) distribution of steps sizes and that $\Delta pilK$ has largely lost the sharp-shoulder peaks in the forward and backward directions (Fig 2C, right, short blue and red regions, respectively). However, unique to $\Delta pilK$ are peaks observed at larger displacements of 1.56 ± 0.06 μm (Fig 2C, right, blue and red regions, and S3 Fig, triangles). These big peak steps are maintained across all time durations, with step sizes slightly larger in the parallel positive direction than parallel negative, yet step sizes in both directions are greater than those for $\Delta chpB$ (Fig 2D, right, and S3 Fig, middle row). Taken together, these analyses suggest that $\Delta pilK$ shares features with $\Delta chpB$ such as reduced frequency of large steps that culminate in failure to bias motility direction but differs with marginally larger and asymmetric parallel and antiparallel steps when taken. These modestly longer steps in the parallel direction, which are absent in the perpendicular direction, may explain why groups of $\Delta pilK$ can macroscopically maintain some seemingly directional motility towards *S. aureus* secreted peptides but additionally has a slightly larger motility zone going away from the peptide gradient, while $\Delta chpB$ does not, since cells tend to maintain directional persistence in the parallel direction (Fig 1E).

Since PilK and ChpB are predicted to modify PilJ methylation status, the bimodal nature of $\Delta pilK$ motility and phenotype of $\Delta chpB$ suggest that *P. aeruginosa* cells depend on PilJ signaling for proper control of step size to orchestrate movements up increasing chemoattractant gradients in the environment. While inactivation of ChpB or PilK can increase step size and lead to enhanced displacement, the displacements are not always directed towards the sensed signals. Extreme changes to pilus activity also appear to exaggerate the pauses in twitching motility, which may disrupt signal sensing by PilJ, as the cells are not as consistently moving through the gradient. For $\Delta pilK$, low pilus activity may result in loss of any pilus dynamic control, potentially explaining why these cells are more susceptible to minute changes in environmental conditions, which leads either to the inability to move or uncontrollable and sporadic movements away from the microcolony. Further investigation is required to resolve how these mutations disrupt downstream signaling and control of pilus dynamics. Regardless, $\Delta chpB$ and $\Delta pilK$ demonstrate failure to coordinate biased steps for chemotaxis towards interspecies signals and support the model that the chemoreceptor PilJ is a necessary component of pilus-mediated chemotaxis.

We previously reported that PilJ was not necessary for *P. aeruginosa* to bias movement towards *S. aureus* [3]. Since the current data reveal a role for 2 enzymes predicted to modify PilJ, we revisited the necessity of PilJ in interspecies signaling here. One hypothesis for the prior observations was that motile $\Delta pilJ$ cells were using flagella-mediated motility, which

obscured the pilus-mediated defect. To test this hypothesis, we generated a mutant lacking both *pilJ* and *flgK*, the flagellar hook; therefore, this mutant was not able to use flagella-mediated motility [40]. Live-imaging shows that Δ*pilJ* Δ*flgK* cells are nonmotile and, thus, fully nonresponsive to *S. aureus*, supporting the hypothesis that flagella-mediated motility had interfered with evaluation of pilus-mediated response to *S. aureus* (S9 Video). Additionally, as shown prior with a Δ*pilA* Δ*flgK* mutant, which similarly lacked functional pili and flagella, Δ*pilJ* Δ*flgK* cells are unable to remain in a clustered microcolony at later time points (S9 Video) [3]. To test that Δ*chpB* and Δ*pilK* phenotypes at the single-cell level were not due to flagella-mediated motility as well, we generated Δ*chpB* Δ*flgK* and Δ*pilK* Δ*flgK* mutants and live-imaged each in monoculture. Both of these mutants phenocopy the behaviors of their respective parental single mutant strains (S10–S12 Videos). Lastly, to confirm that the altered motility phenotypes of the methyl modification mutants are not due strictly to a change in the amount of cellular PilJ, a C-terminal His tag was fused to PilJ at the native chromosomal site and PilJ levels were detected in protein lysates from wild type, Δ*chpB*, and Δ*pilK* mutants by western blot (S6 Fig). PilJ levels were stable and similar in all strains, supporting the hypothesis that changes in protein expression of the chemoreceptor are not responsible for the loss in bias of directional movement, despite increases in single-cell motility (S6 Fig).

## Results: Methyl modification of PilJ is necessary for TFP-mediated chemotaxis response to *S. aureus*

The necessity of PilK and ChpB for directional twitching motility suggests that, like other bacterial chemoreceptors, methylation levels of PilJ influence the downstream signaling cascade. We hypothesized that PilJ contains at least one methylation site necessary for chemoreceptor adaptation and that, in comparison to a *pilJ* deletion, cells harboring mutations in PilJ methylation motifs would produce functional protein and remain motile, providing a valuable tool to monitor the role of PilJ in *P. aeruginosa* interspecies chemotaxis. Since, the number and location of the methylation sites in PilJ are unknown, we first searched the amino acid sequence of PilJ for the conserved 10 amino acid methylation motif. This motif, [A/S/T/G]-[A/S/T/G]-X-X-[**E/Q**]-[**E/Q**]-X-X-[A/ S/T/G]-[A/S/T/G], has a pair of glutamine and/or glutamate residues at the center [41–44]. In other MCPs, one of these residues is methylated by the methyltransferase and demethylated by the methylesterase. Examination of PilJ reveals 2 motifs with an exact match to the conserved sequence and 1 motif that shares 9 of the 10 conserved amino acids. This suggests PilJ has 3 potential methylation sites in the predicted cytoplasmic region (Fig 3A, black and pale orange circles, and S7 Fig). The predicted methylation sites are located at residues Q412/E413, Q623/Q624, and Q639/E640.

To experimentally determine whether the methylation sites were necessary for TFP-mediated chemotaxis, we next generated a mutation in each methylation site by substituting the glutamine/glutamate residue pair to an alanine/alanine pair. Given most other *pilJ* mutants exhibit severely diminished twitching motility, we first compared the ability of each PilJ methylation site mutant to twitch in a standard subsurface motility assay [45]. Briefly, *P. aeruginosa* was inoculated at the plastic-agar interface and allowed to move prior to measuring the diameters of motility. Only the *pilJ*$_{Q412A, E413A}$ mutant retains full wild type levels of motility, while *pilJ*$_{Q623A, Q624A}$ has moderate-but-significant reduction in motility and *pilJ*$_{Q639A, E640A}$ is severely diminished in twitching motility (Fig 3B). We next asked if mutation of these predicted methylation sites alters *P. aeruginosa* attraction towards *S. aureus*. In the macroscopic directional motility assay, *pilJ*$_{Q412A, E413A}$ demonstrates significant loss of migration up the gradient of *S. aureus* supernatant (Fig 3C). The other methylation site mutants also display attenuated chemoattraction to *S. aureus*; yet, it is unclear how much of this reduced response is due

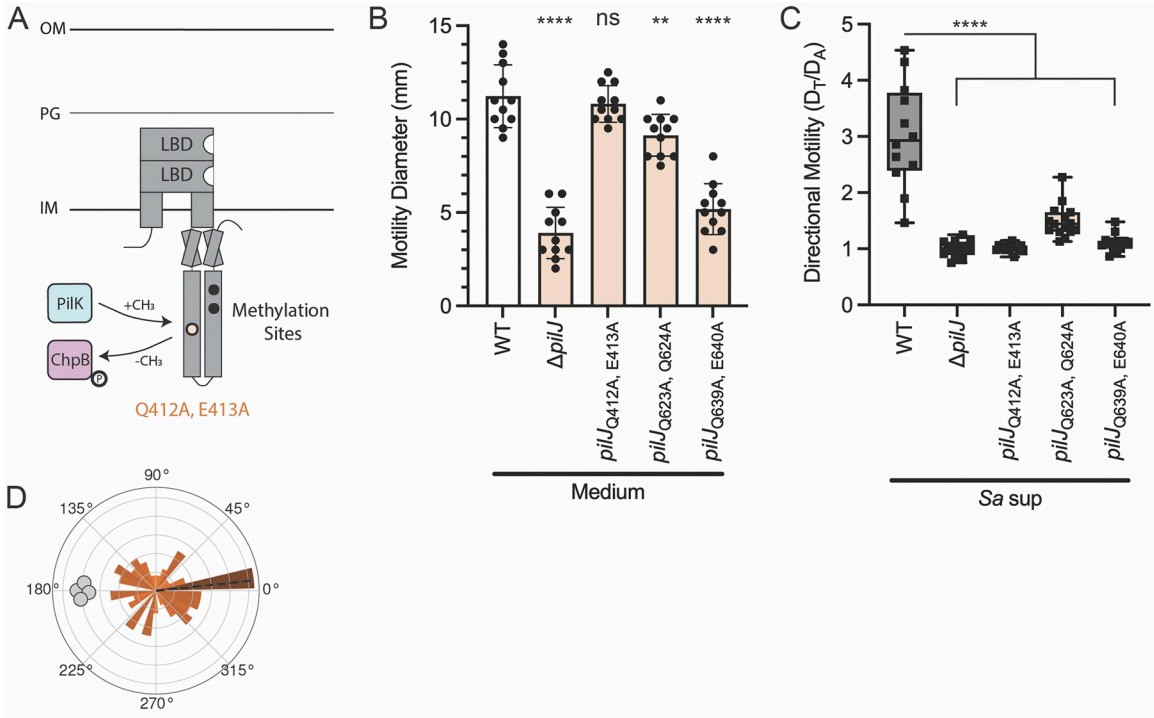

**Fig 3. Methyl modification of PilJ is necessary for TFP-mediated chemotaxis response to *S. aureus*.** (**A**) Schematic of PilJ with cytoplasmic methylation sites represented as black circles. The methylation site Q412, E413, whose mutant retains wild type motility and is further studied for chemotaxis response, is highlighted in pale orange. (**B**) Twitching motility diameters of *P. aeruginosa* wild type and methylation mutants. Motility diameters were analyzed by a one-way ANOVA followed by Dunnett's multiple comparisons test. **** indicates $p < 0.0001$; ** indicates $p < 0.01$; *ns* indicates no statistically significant difference. (**C**) Migration towards *S. aureus* secreted factors of *P. aeruginosa* wild type and methylation mutants *pilJ*$_{Q412A, E413A}$, *pilJ*$_{Q623A, Q624A}$, and *pilJ*$_{Q639A, E640A}$. Statistical significance of directional motility was determined with a one-way ANOVA followed by Dunnett's multiple comparisons test. **** indicates $p < 0.0001$. Motility diameters and directional motility are shown for at least 3 biological replicates each containing a minimum of 3 technical replicates. (**D**) Rose graph of the principal angle of motility for each cell trajectory relative to starting position for *P. aeruginosa* *pilJ*$_{Q412A, E413A}$ in coculture with *S. aureus*. Position of *S. aureus* relative to the center of the *pilJ*$_{Q412A, E413A}$ cells is represented by the gray cocci on the perimeter. Trajectory angles are shown by colored vectors with the average angle of all trajectories represented by the dotted black line. Larger vectors indicate more cells for the given principal angle. One rose graph representative of at least 3 replicates is shown. The underlying data can be found in S1 and S5 Data. IM, inner membrane; LBD, ligand binding domain; OM, outer membrane; PG, peptidoglycan; TFP, type IV pilus; WT, wild type.

to motility defects versus deficiency in pilus-mediated chemotaxis; therefore, we focused on *pilJ*$_{Q412A, E413A}$ moving forward (Fig 3C). To confirm that the migration phenotype of *pilJ*$_{Q412A, E413A}$ can be restored with full-length *pilJ*, the mutant and wild type strains were complemented with a plasmid containing a GFP-tagged copy of wild type *pilJ* under control of an arabinose-inducible promoter (S8 Fig). This allows for visualization of PilJ in the cells, which show the expected bipolar localization (S8A Fig). Additionally, the complemented mutant strain shows levels of directional response similar to wild type harboring the complementation plasmid (S8B Fig). Due to the GFP tag, some PilJ signaling may be diminished leading to the lower interspecies signal response in the complemented strains.

Next, we live-imaged the *pilJ*$_{Q412A, E413A}$ mutant in monoculture and observe that it moves earlier than wild type, Δ*pilK*, or Δ*chpB*, with single-cell or small groups of 2 to 3 cells traveling together (S13 Video). Furthermore, *pilJ*$_{Q412A, E413A}$ does not form a microcolony; rather, the cells begin to twitch and move apart from each other at very low cell density, typically before there are approximately 10 cells present. This behavior is recapitulated in the presence of *S. aureus*, with *pilJ*$_{Q412A, E413A}$ additionally showing no bias in movement towards *S. aureus*

(Fig 3D and S14 Video). The *pilJ*$_{Q412A, E413A}$ mutant cells also become elongated after a few hours compared to wild type. This slight cell division defect is likely due to the high cAMP levels in *pilJ*$_{Q412A, E413A}$ (see Fig 5). Because of this possibility and the increased twitching motility exhibited by *pilJ*$_{Q412A, E413A}$, protein levels of PilJ$_{Q412A, E413A}$ were evaluated by purification of His-tagged wild type PilJ and PilJ$_{Q412A, E413A}$ followed by western blot (S6 Fig). This analysis revealed that both PilJ proteins are expressed at similar levels, which supports that the hyper-motile phenotype of the *pilJ*$_{Q412A, E413A}$ mutant is not due to an excess of mutant PilJ in the cells and that the mutation to methylation site does not impact PilJ stability (S6 Fig). Live-imaging of a *pilJ*$_{Q412A, E413A}$ Δ*flgK* mutant in coculture with *S. aureus* also phenocopies the parental *pilJ*$_{Q412A, E413A}$ (S15 Video). Together, these data predict 3 glutamate and/or gluta-mine methylation sites in the cytoplasmic domain of PilJ and show that 1 site is required for chemotaxis, but not the ability to twitch, suggesting that methyl modification of at least 1 site is essential for Pil-Chp chemotaxis signaling. Meanwhile, the other 2 sites are necessary for full twitching motility; therefore, dissection of the role for each site in chemotaxis alone could not be assessed.

## Results: The ligand binding domains of PilJ are required for pilus-mediated chemotaxis but not twitching motility

Given methylation adaptation is necessary to bias movement towards *S. aureus* but not twitch-ing, we next asked if the LBDs of *P. aeruginosa* PilJ are also required for chemotaxis. We gener-ated a mutant lacking the periplasmic portion containing both predicted PilJ LBDs (residues 39-303), based on the predictions by Martín-Mora and colleagues and confirmed by Alpha-Fold, yet kept the transmembrane domains and entire cytoplasmic region intact, which we now refer to as *pilJ*$_{ΔLBD1-2}$ (Fig 4A, LBDs in pale orange) [46]. To ensure that PilJ$_{ΔLBD1-2}$ is sta-ble, western blots with His-tagged PilJ$_{ΔLBD1-2}$ were performed (S6 Fig). Like the wild type, PilJ$_{ΔLBD1-2}$ is well expressed and exhibits a protein band at the expected smaller size around 46 kDa compared to the wild type protein band near 75 kDa (S6 Fig).

We then tested whether *pilJ*$_{ΔLBD1-2}$ retained any twitching motility using the subsurface twitching assay, as described above, and found that loss of the LBDs does not reduce the ability to twitch (Fig 4B). Since *pilJ*$_{ΔLBD1-2}$ is able to twitch to wild type levels, we next tested the extent that it could chemotax up a gradient of *S. aureus* secreted factors. Despite being motile, *pilJ*$_{ΔLBD1-2}$ loses nearly all ability to move directionally towards *S. aureus* secreted factors (Fig 4C). *P. aeruginosa* wild type and *pilJ*$_{ΔLBD1-2}$ chemotaxis were further evaluated in the presence of a gradient of supernatant derived from a *S. aureus* PSM mutant. Neither the wild type nor *pilJ*$_{ΔLBD1-2}$ strain shows directional motility response to the gradient of secreted factors lacking PSMs (Fig 4D). When complemented with *pilJ-gfp*, *pilJ*$_{ΔLBD1-2}$ exhibits bipolar localization of wild type PilJ; however, only partial restoration of interspecies signal response is observed, despite statistical similarity to wild type with the complement plasmid (S9 Fig). Chemorecep-tors are typically grouped in arrays of trimer-of-dimer units with each unit signaling down-stream to a kinase [36]. Therefore, despite proper localization of the wild type PilJ, the combination of PilJ$_{ΔLBD1-2}$ mutant and GFP-tagged wild type copies of PilJ for this particular strain may lead to trimers-of-dimers with inefficient signaling for complete directional response (that is, PilJ$_{ΔLBD1-2}$ functions as a dominant negative to inhibit function of the wild type). Finally, we determined if the PilJ LBDs are necessary for *P. aeruginosa* chemotaxis up a gradient of PSMs and found, similar to the chemoattraction towards *S. aureus* wild type super-natant, *P. aeruginosa* wild type is attracted to δ-toxin, in a dose-dependent manner, while *pilJ*$_{ΔLBD1-2}$ is blind to this gradient at all concentrations tested (Fig 4D). These data further demonstrate that PSMs are a secreted chemoattractant sensed by *P. aeruginosa*, that the LBDs

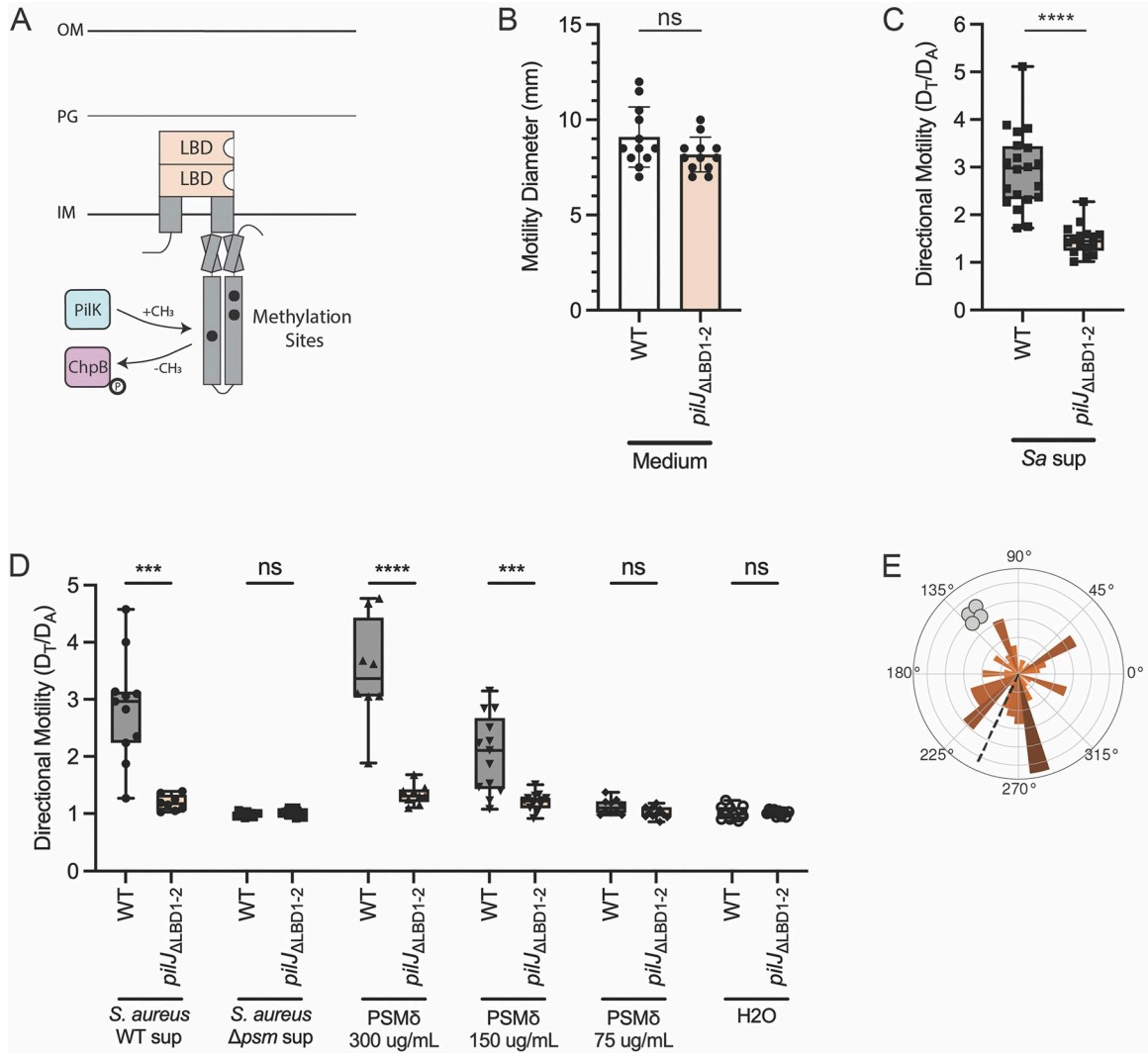

**Fig 4. The ligand binding domains of PilJ are required for pilus-mediated chemotaxis but not twitching motility.** (**A**) Schematic of PilJ with periplasmic LBDs highlighted in pale orange. Twitching motility diameters (**B**) and directional motility towards *S. aureus* secreted factors (**C**) or *S. aureus* PSM δ-toxin at multiple concentrations (**D**) of *P. aeruginosa* wild type and *pilJ*$_{\Delta LBD1-2}$. Macroscopic motility measurements were analyzed with an unpaired *t* test. **** indicates *p* < 0.0001; *ns* indicates no statistically significant difference. Motility diameters and directional motility are shown for at least 3 biological replicates each containing a minimum of 3 technical replicates. (**E**) Rose graph of the principal angle of motility for each cell trajectory relative to starting position for *P. aeruginosa* *pilJ*$_{\Delta LBD1-2}$ in coculture with *S. aureus*. Position of *S. aureus* relative to the center of the *pilJ*$_{\Delta LBD1-2}$ cells is represented by the gray cocci on the perimeter. Trajectory angles are shown by colored vectors with the average angle of all trajectories represented by the dotted black line. Larger vectors indicate more cells for the given principal angle. One rose graph representative of at least 3 replicates is shown. The underlying data can be found in S1 and S6 Data. IM, inner membrane; LBD, ligand binding domain; OM, outer membrane; PG, peptidoglycan; PSM, phenol soluble modulin; WT, wild type.

of PilJ are necessary to sense PSMs, and that the sensing of PSMs leads to directed motility response.

Following examination of community-level movement and response, we performed live-imaging of *pilJ*$_{\Delta LBD1-2}$ in monoculture to determine how loss of the LBDs impacted single-cell movements. The *pilJ*$_{\Delta LBD1-2}$ mutant displays increased and earlier motility, with cells lacking microcolony formation seen by the wild type (S16 Video). Unlike the methylation PilJ mutant, *pilJ*$_{\Delta LBD1-2}$ tends to move as groups of cells, with trajectories curving in wide loops

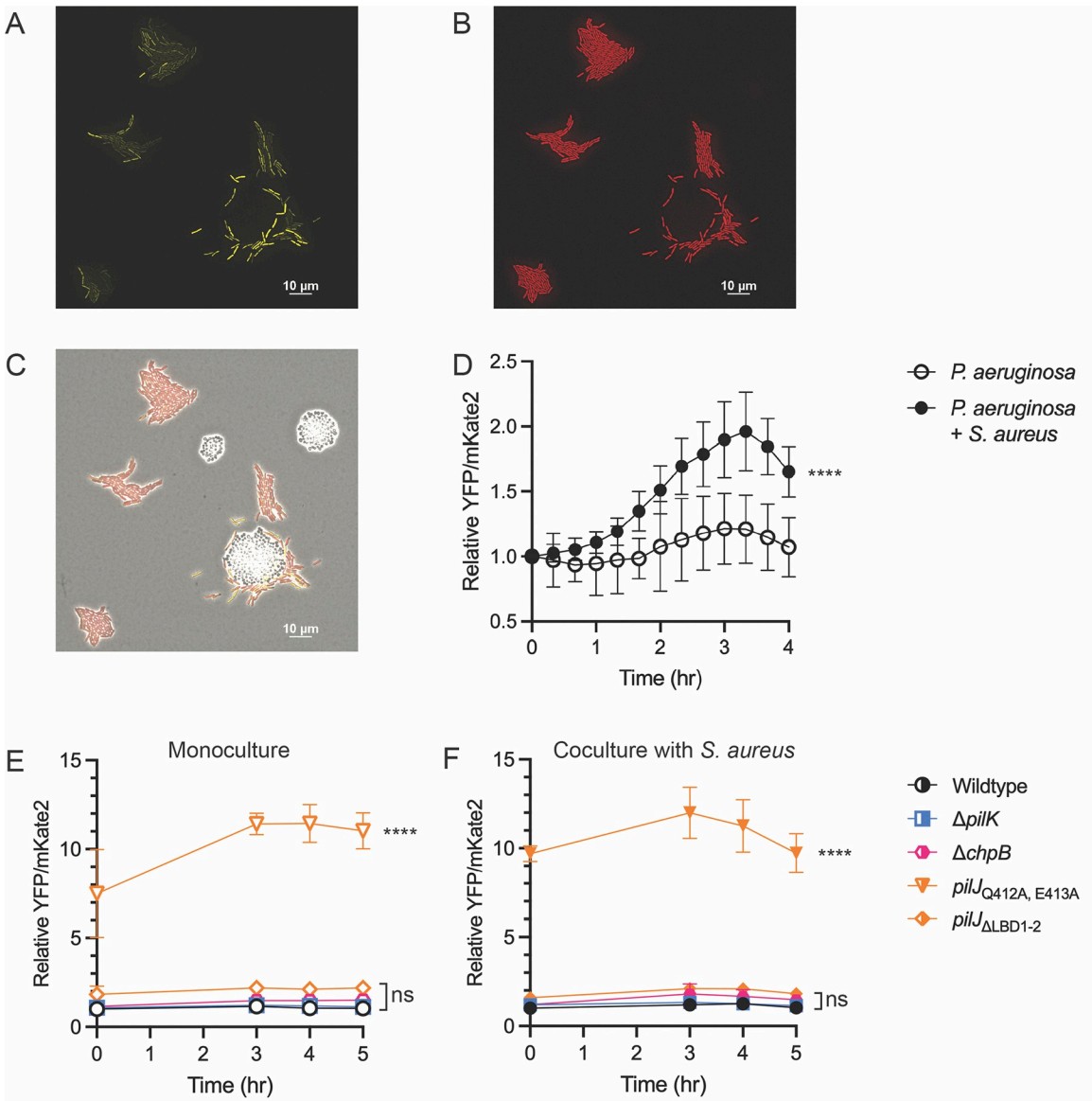

**Fig 5. cAMP changes during pilus-mediated chemotaxis independent of TFP activity.** (**A–D**) Intracellular levels of *P. aeruginosa* cAMP measured in monoculture and coculture with *S. aureus* using a *P. aeruginosa* PAO1 strain carrying a reporter with the cAMP-responsive promoter P$_{xphA}$ transcriptionally fused to *yfp* and constitutively expressed promoter P$_{rpoD}$ fused to *mKate2* for normalization. Representative coculture images at t = 3.5 hours for the YFP (**A**), TxRed (**B**), and merged channels (**C**). cAMP levels of *P. aeruginosa* PA14 *pil-chp* mutants in monoculture (**E**) and coculture with *S. aureus* (**F**) using the cAMP reporter. cAMP levels were monitored by dividing YFP by mKate2 fluorescence intensity for each time point and normalizing intensity to wild type cAMP at t = 0 hour. cAMP levels are shown for at least 4 microcolonies per condition and were compared using either multiple unpaired *t* tests or a one-way ANOVA followed by Dunnett's multiple comparison's test. **** indicates *p* < 0.0001; *ns* indicates no statistically significant difference. The underlying data can be found in S1 Data. cAMP, cyclic adenosine monophosphate; TFP, type IV pilus.

(S16 Video). These behaviors are echoed in coculture with *S. aureus*, with *P. aeruginosa* *pilJ*$_{\Delta LBD1-2}$ appearing to show no bias towards *S. aureus* microcolonies (Fig 4E and S17 Video). This behavior is also phenocopied by a *pilJ*$_{\Delta LBD1-2}$ Δ*flgK* mutant (S18 Video). Collectively, these data establish a role for the LBDs of PilJ in control of response to interspecies signals, while also showing they are not essential for general pilus-mediated motility.

## Results: cAMP changes during pilus-mediated chemotaxis independent of TFP activity

Intracellular levels of the second messenger cAMP and Pil-Chp activity are connected. As signal transduction through Pil-Chp increases, the response regulator PilG activates the adenylate cyclase, CyaB, necessary for cAMP synthesis [25]. In turn, cAMP indirectly increases transcriptional expression of *pil-chp* generating a positive feedback loop between the 2 systems [25,27].

Due to the role of Pil-Chp in perception and reaction to *S. aureus*, we asked whether *P. aeruginosa* increased cAMP levels during interactions with *S. aureus*. To answer this question, we live-imaged a previously established cAMP reporter strain *P. aeruginosa* wild type PAO1 carrying a cAMP-responsive promoter, $P_{xphA}$, fused to *yfp* and a constitutively expressed promoter $P_{rpoD}$, fused to *mKate2* [26]. Kinetic cAMP levels were measured in individual cells in monoculture or coculture with *S. aureus*, with YFP normalized to mKate2 fluorescence. cAMP is known to be heterogenous among cells in a population. In wild type *P. aeruginosa* monoculture, this heterogeneity is observed but with minor changes in cAMP levels over time. However, in coculture, cAMP increases, particularly in cells that move towards and surround *S. aureus* (Fig 5A–5D). This suggests that response to *S. aureus* is associated with increases in cAMP. Furthermore, when community-level response of *P. aeruginosa* lacking either CyaB or the cAMP phosphodiesterase, CpdA, was examined, *P. aeruginosa* cannot fully move up a gradient of *S. aureus* supernatant, despite retaining twitching motility (S11 Fig). These data suggest that *P. aeruginosa* modulation of cAMP levels is necessary for proper TFP-mediated response to interspecies signals.

Next, we asked how cAMP levels compared between response-deficient mutants and the parental *P. aeruginosa* (PA14). While reporter levels in PA14 are lower in comparison to PAO1, a similar trend is observed. Each mutant shows at least some increase in cAMP relative to wild type cAMP levels in monoculture or coculture, though not significant for Δ*pilK* and Δ*chpB*. The *pilJ*$_{ΔLBD1-2}$ mutant exhibits an approximately 2-fold increase in cAMP over time compared to wild type, whereas, *pilJ*$_{Q412A, E413A}$ exhibits a 10-fold increase, independent of the presence of *S. aureus* (Fig 5E and 5F). These data suggest that cAMP levels are influenced by chemoreceptor-mediated signaling. However, the degree to which cAMP increases does not directly correlate with the extent of increased motility or aberrant chemosensory response, supporting the model that spatial and temporal precision of cAMP signaling is necessary for effective TFP response to chemotactic stimuli and highlighting the complexity of the Pil-Chp/cAMP signaling network.

## Discussion

While the Pil-Chp system has been studied for its roles in twitching motility and cAMP regulation, less work has explored the additional roles of the system in TFP-mediated chemotaxis. Furthermore, most studies have primarily investigated PilJ through a nonmotile complete-deletion mutant, which allowed little understanding of the domains of PilJ that determine chemoreceptor activity. Here, we utilized domain-specific mutations of PilJ to evaluate single-cell and community-level behaviors that define the importance of PilJ to sense and relay interspecies signals to pilus response regulators. These observations show *P. aeruginosa* is able to move towards interspecies signals through a novel TFP-mediated chemotaxis mechanism and that PilJ does have the necessary components to serve as a MCP for signal sensation, transmission, and adaptation (Fig 6). To our knowledge, this is the first study to generate mutants in the methylation sites or lacking a majority of both LBDs of PilJ, which retain wild type levels of motility and define their contribution to chemotactic regulation.

While flagella-mediated chemotaxis of swimming bacteria has been extensively characterized, there is comparatively little insight into bacterial chemotaxis on a surface. For decades, it has been predicted that systems like Pil-Chp may have chemotactic functions, given the protein homology to established flagella chemotaxis system components [33]; yet, for many of these proteins, the homology is low and some proteins consist of several more domains relative to their flagella-mediated chemotaxis counterparts, like the Pil-Chp histidine kinase ChpA, which has 8 Hpt/Spt/Tpt domains for phosphorylation compared the single Hpt domain of Che histidine kinase, CheA [35,47]. This suggests that while chemosensory systems maintain broad similarities, there are fundamental differences between the mechanism of pili and flagella-driven chemotaxis.

During swimming motility, when the flagellar motor rotates counterclockwise (CCW), cells perform smooth "runs" in a persistent direction. Periodically, the flagellar motor will switch rotational direction to clockwise (CW), producing a tumble in peritrichous bacteria (such as *Escherichia coli*) or a pause and turn, as seen in the monotrichous *P. aeruginosa*. CW rotation occurs when the CheA kinase phosphorylates the response regulator CheY, which engages with the motor proteins to induce conformational changes that lead to a shift in the direction of flagellar rotation. Following a tumble or turn, the cell body randomly pivots in a different direction. Dephosphorylation of CheY later leads to disengagement from the motor, thereby returning the motor to the CCW rotational conformational state that results in a run in the new direction. By biasing the rotation of the motor more frequently in the CCW direction when moving towards an increasing concentration of attractant, the cells perform longer runs and, thus, bias movement up the gradient.

By analogy, current models for type IV pili chemotaxis predict that directionality is driven by the frequency that cells perform "runs" and "reversals." Reversals are thought to occur when the cell switches which pole the pili are extended and retracted from. For example, if cells are experiencing an increasing concentration of chemoattractant, the frequency of reversals would be lower, allowing cells to maintain persistent movement through the gradient. The Pil-Chp system contains 2 CheY-like response regulators, PilG and PilH. While both response regulators are thought to influence the activity of the pilus extension and retraction ATPases, respectively, it remains unclear which, or both, may be necessary for control of reversals and the mechanisms regulating activation.

Recent data suggest that, under nonchemotactic conditions, phosphorylation of PilG by the kinase ChpA drives asymmetric polar localization of PilG and pilus activity to the leading pole of motion [18]. Furthermore, Oliveira and colleagues demonstrated cells lacking *pilG* are unable to bias motility up a gradient of DMSO and performed fewer reversals than wild type cells [2]. This investigation concluded that *P. aeruginosa* adopts a "pessimistic" strategy for pilus-mediated chemotaxis, in which cells tend to respond through increased reversals when conditions get worse, that is, when chemoattractant concentrations decrease [2]. However, it is still unclear how PilG increases reversals in wild type cells and what the influence of PilH in bias and reversals may be.

Although we were unable to measure reversals under our conditions, the data presented here highlight step size and frequency as determinants of bias in pilus-mediated chemotaxis but do not eliminate a role for reversals in twitching chemotaxis (Fig 2). From Kuhn and colleagues and Oliveira and colleagues, we see that PilG, and perhaps PilH, seem to generate increased pilus activity at the leading pole to control reversals necessary for biased movements. If we follow the pessimistic chemotaxis strategy, then one possibility is that *P. aeruginosa* twitching cells generally take uniform steps as they sample the environment; yet, when steps down the gradient are taken, PilG and PilH are activated or shuttled to the leading pole to both reverse direction and increase pilus activity that results in larger steps up when

chemoattractant concentration increases (Fig 2). At this time, PilJ would also be expected to be reset by the response regulator and methylesterase ChpB. We hypothesize that the demethylation of PilJ changes conformation in a way that alters the phosphorylation of PilG and PilH but still do not understand the exact relationship between the chemoreceptor and response regulators or the outcome of chemoreceptor adaptation on step sizes. Furthermore, whether PilG and PilH work together at the leading pole, the Pil-Chp targets and their spatiotemporal activation/phosphorylation are unknown. Future studies are required to fully understand the roles of the response regulators in type IV pilus chemotaxis.

Despite overall sequence similarity of PilJ in the cytoplasmic region to other *P. aeruginosa* MCPs, the low similarity of the periplasmic region containing the LBDs has led to several questions about the function of this putative pilus MCP. First, what ligands does PilJ bind? Martín-Mora and colleagues previously investigated the only other *P. aeruginosa* MCP containing a PilJ LBD, McpN, which was shown to bind nitrate [46]. Evaluation of the McpN and PilJ ligand binding pocket motifs showed little conservation between the 2 MCPs and PilJ lacked nitrate binding ability [46]. If PilJ does not bind nitrate, then what signals can it bind? Persat and colleagues showed that PilJ can interact with the pilus monomer PilA in the periplasmic regions of each protein and proposed this interaction regulates mechanosensing [26], although recent data argue against this model [48]. However, it is possible that PilJ holds multiple but distinct roles as both a conventional chemoreceptor and a sensor for PilA levels in the inner membrane. *P. aeruginosa* has a second nonconventional chemosensory system for surface sensing, called Wsp. Recently, it was shown that the Wsp system is more broadly a membrane stress detection system and surfaces are just one of many membrane stressors that the receptor WspA detects [49]. In eukaryotic cells, PSMs are known to form a membrane-perturbing pore; thus, PilJ may also perceive and transduce interspecies peptide-induced membrane stress (Fig 6) [50].

If PilJ binds a conventional chemoattractant signal, full-length PSMs are unlikely to interact directly with the LBD, due to their size and amphipathic secondary structure. Nolan and colleagues identified that *P. aeruginosa* exhibits increased twitching in the presence of environmental signals, such as tryptone, mucins, or bovine serum albumin [51]. This response required *P. aeruginosa* protease activity, presumably to cleave environmental factors into smaller signals [51]. While this group did not study the chemotactic nature of these compounds, they do suggest that the increased twitching response is PilJ-mediated. This indicates that PilJ may sense a broad range of environmental signals. It is thus plausible that the *S. aureus* peptides, which are shown to elicit a PilJ-dependent chemotaxis response here, are indeed a chemoattractant signal. Yet, it is still unclear how PSMs may gain access to and bind the periplasmic LBDs of PilJ and thus act as a traditional chemoattractant signal for activation of downstream signaling. Therefore, protease-cleavage of interspecies peptides may be necessary to fragment PSMs into signal-sized peptides for either direct or indirect PilJ binding, in conjunction with earlier observations that *P. aeruginosa* chemotaxes towards di- or tripeptides rather than larger oligopeptides [52]. Another possibility is that the peptides bind PilJ indirectly, perhaps through a solute binding protein, as previously reported for chemoattractant inorganic phosphate indirectly binding chemoreceptor CtpL through the mediating solute binding protein PtsS [28,53]. Investigation of these hypotheses, including a role for membrane stress, is currently underway (Fig 6).

Jansari and colleagues previously reported that their *pilJ* mutant retained some twitching motility but had diminished response to phosphatidylethanolamine [54]. In our current study, chemoattraction to phosphatidylethanolamine by wild type *P. aeruginosa* was not observed, and, therefore, chemotaxis towards phosphatidylethanolamine by the *pilJ* mutants generated here could not be determined. Jansari and colleagues generated a *pilJ* mutant lacking residues

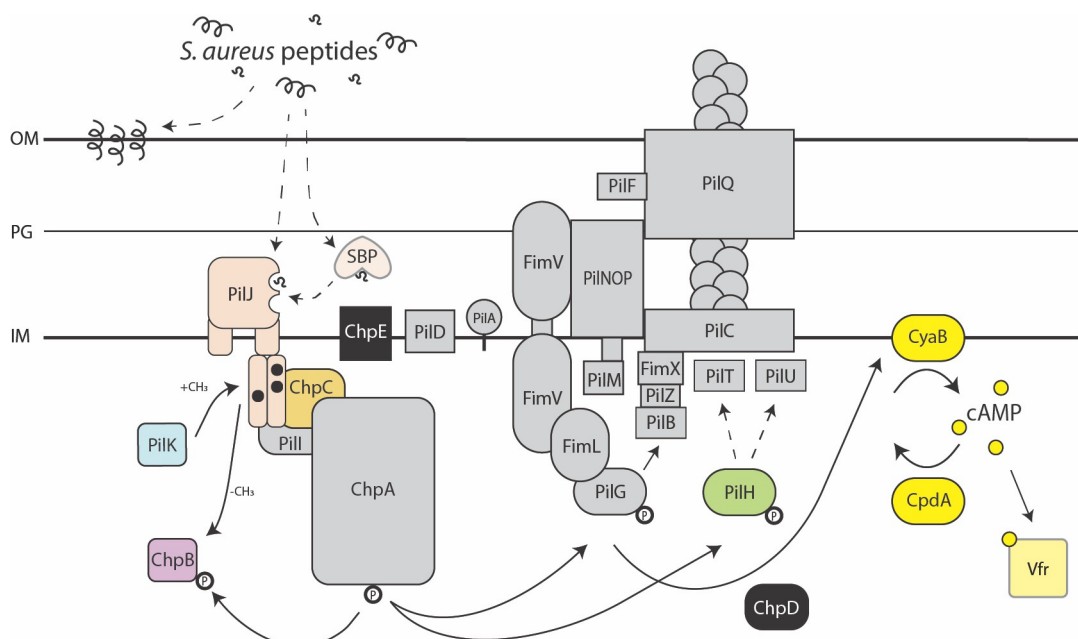

**Fig 6. Model of TFP-mediated interspecies chemotaxis.** *S. aureus* peptide signals (black) are sensed by *P. aeruginosa* Pil-Chp chemoreceptor PilJ (pale orange) LBDs and transmitted to histidine kinase ChpA (gray), likely through the linker protein ChpC (gold). The kinase state modulates the activity of response regulators PilG (gray) and PilH (green), which coordinate TFP extension and retraction events that bias movements towards the interspecies signals. Adaptation of PilJ through modification of the methylation sites by enzymes, PilK (blue) and ChpB (purple), is also required for proper chemotactic response. Chemotaxis through the Pil-Chp system also leads to increases in cAMP (yellow) and depends on careful regulation of these cAMP levels by enzymes CyaB and CpdA (yellow). cAMP activates Vfr for up-regulation of virulence factors, including the Pil-Chp system. The chemoattractant peptide signals sensed by PilJ may be full-length *S. aureus* PSMs (black helices) or oligopeptide cleavage products (black squiggles). Additionally, these peptides, whether whole or cleaved, could directly bind PilJ or indirectly activate PilJ via an interaction with a periplasmic SBP. Alternatively, *S. aureus* peptides may embed into the *P. aeruginosa* membrane, causing transient membrane stress and activating Pil-Chp. The solid arrows indicate previously described interactions and the dashed arrows indicate hypothetical pathways that will be tested in future studies. cAMP, cyclic adenosine monophosphate; IM, inner membrane; LBD, ligand binding domain; OM, outer membrane; PG, peptidoglycan; SBP, solute binding protein; TFP, type IV pilus.

74-273, a portion of the prediction LBDs [54]. Examination of a similar mutant lacking residues 80-273 in the present investigation shows that *P. aeruginosa* pilJ$_{\Delta 80-273}$ similarly has significant reduction in twitching; yet, this defect unfortunately yields insufficient motility to measure reduced attraction to *S. aureus* and, thus, could not be evaluated here (S10 Fig). However, the mutant lacking both PilJ domains (residues 39-303; pilJ$_{\Delta LBD1-2}$) retains near wild type levels of motility and allows for visualization of response deficiency at both the community and single-cell levels. These observations show that without the PilJ LBDs, *P. aeruginosa* is unable to bias motility towards interspecies signals, which further supports that the periplasmic portion of PilJ is not essential for twitching motility but is important for sensing signals—whether they are chemoattractants or surface signals. This suggests PilJ has evolved to coordinate TFP motility in response to several environmental factors. It remains unknown, however, why *P. aeruginosa* PilJ contains 2 pilJ LBDs and whether they bind ligands cooperatively or independently, or if each has a designated role for particular chemo- or surface-sensing signals.

Little was previously known about the methylation sites on PilJ, which are typically required for chemotaxis adaptation. While bacterial chemoreceptors have a conserved motif for methylation sites, potential motifs on PilJ had not been identified. To our knowledge, this work is the first to identify 3 likely residue pairs for methyl modification on PilJ, further characterizing the

chemoreceptor. In other bacterial chemoreceptors, mutation of each methylation site on a chemoreceptor does not lead to the same phenotype [55]. Thus, it is reasonable that mutations in each PilJ methylation site differentially influence the conformation of PilJ signaling domains and lead to the 3 different macroscopic motility behaviors described here (Figs 3B and S8). While live-imaging was only performed on the first methylation site, this was sufficient to show changes in these sites can dramatically alter *P. aeruginosa* chemotaxis (S13 Video).

Increases in *P. aeruginosa* cAMP are commonly associated with surface sensing and Pil-Chp activity. It is further established that while PilG activates CyaB for cAMP production, cAMP in turn increases expression of *pil-chp* genes [25,27]. However, previous reports focused on Pil-Chp activity in terms of twitching motility; thus, a link between chemotactic Pil-Chp activity and cAMP levels had been ambiguous. While each mutant shown here has varying levels of motility, all are increased in both motility and cAMP levels relative to wild type; yet, there is not a direct correlation between mutants that have increased motility and the degree to which cAMP is increased. Only $pilJ_{Q412A, E413A}$ showed much higher levels of cAMP at nearly 10-fold the amount as wild type. While the precise role of cAMP signaling in chemotaxis is unclear, studies are currently underway to further interrogate this signaling pathway.

Dissection of Pil-Chp and its role in chemotaxis towards interspecies signals has broadened understanding of a unique bacterial chemosensory system that may be utilized for bacterial communication and survival in complex, polymicrobial environments. During *P. aeruginosa–S. aureus* coinfections, such as those in cystic fibrosis airways, patients often succumb to worse clinical outcomes than their counterparts who are only infected by one organism [4]. Furthermore, once coinfected, patients tend to stay infected by both organisms for several years [56]. Such stable, long-term polymicrobial infections may be enhanced by the chemical and physical interactions between species seen here. With this knowledge of how bacteria can sense their respective secreted factors, new therapeutic strategies targeting this system may provide the opportunity to break communication between species and prevent these detrimental interactions, thereby eliminating infections and consequently improve patient outcomes.

## Methods and materials

### Bacterial strains and culture conditions

*P. aeruginosa* PA14 or PAO1 and *S. aureus* JE2 strains were cultured in tryptic soy broth (TSB; Becton Dickenson) or M8 minimal media broth supplemented with 0.2% glucose and 1.2% tryptone (M8T) with aeration at 37°C. The following antibiotics were added for *P. aeruginosa* cultures when appropriate: carbenicillin (200 μg/mL), gentamicin (30 μg/mL), and tetracycline (100 μg/mL). *E. coli* strains for cloning were cultured in lysogeny broth (LB; 1% tryptone, 0.5% yeast extract, 1% sodium chloride). The following antibiotics were added for *E. coli* when appropriate: ampicillin (100 μg/mL), gentamicin (15 μg/mL), and tetracycline (12 μg/mL). All strains used in this study can be found in S1 Table.

### Generating *P. aeruginosa* mutants

*P. aeruginosa* mutants in this study were generated by allelic exchange at the native site in the chromosome using Gibson Assembly with a pEXG2-Tc vector containing the DNA mutation between restriction sites XbaI and SacI [57]. Mutant and tag constructs were generated by PCR amplifying approximately 1 kb DNA fragments upstream and downstream of the gene or region of interest, while substitution mutants were generated by synthesis of a DNA fragment gene block containing the correct codon change (Integrated DNA Technologies, Coralville, IA, USA). Assembled vectors were transformed into *E. coli* DH5α, then into *E. coli* S17 for conjugation into *P. aeruginosa*. Correct mutations in *P. aeruginosa* were verified with PCR and

Sanger sequencing. *P. aeruginosa* Δ*pilK*, Δ*chpB*, Δ*pilJ* deletion mutants and *pilJ* LBDs and methylation site mutants were complemented by electroporating the respective mutant with expression vector pMQ80 containing the full-length gene under control of the arabinose-inducible $P_{BAD}$ promoter and fused to a C-terminal GFP tag [58]. All oligonucleotides used to generate *P. aeruginosa* mutants can be found in S2 Table.

### Macroscopic coculture twitching chemotaxis assay

Motility experiments were performed as previously described [3,31,59]. Buffered agar plates (recipe: 10 mM Tris (pH 7.6); 8 mM MgSO4; 1 mM NaPO4 (pH 7.6); and 1.5% agar) were poured and allowed to solidify for 1 hour prior to incubation for 16 hours at 37°C and 22% humidity. After solidifying, 4 μL of either growth medium (TSB) or cell-free supernatant derived from an overnight culture of *S. aureus* at $OD_{600}$ 5.0 and filter sterilized with a 0.22-μm filter were spotted on the surface of the plate and allowed to diffuse for 24 hours at 37°C and 22% humidity to establish a gradient. *P. aeruginosa* cultures were incubated overnight in TSB with aeration at 37°C, subcultured 1:100 in TSB the following morning, then standardized to $OD_{600}$ 12.0 in 100 μL of 1 mM MOPS buffer supplemented with 8 mM $MgSO_4$ prior to inoculating 1 μL on the surface of the plate at 5 mm from the center of the gradient. Plates were incubated in a single layer, agar-side down, for 24 hours at 37°C with 22% humidity, followed by an additional 16 hours at room temperature prior to imaging the motility response of *P. aeruginosa*. Images were captured using a Zeiss stereoscope with Zeiss Axiocam 506 camera, and directional motility ratios (S1 Data) were calculated in Fiji before graphing and performing statistical analysis in GraphPad Prism.

### Macroscopic coculture subsurface twitching assay

The assay was modified from a previous protocol, as shown before [3,45]. Prior to pouring plates, 200 μL of either growth medium (TSB) or cell-free supernatant derived from an overnight culture of *S. aureus* at $OD_{600}$ 2.0 and filter sterilized with a 0.22-μm filter were spread on the bottom of a 120-mm square petri plate. Tryptic soy agar (1.5%) was then poured into the plates and allowed to dry for 4.5 hours at room temperature. *P. aeruginosa* cultures were incubated overnight in TSB with aeration at 37°C, subcultured 1:100 in TSB the following morning, then standardized to $OD_{600}$ 2.0 in 1 mL TSB. A sterile toothpick was dipped into the standardized culture, then stabbed to the bottom of the agar plate. Plates were incubated in a single-layer, agar-side down, at 37°C with 22% humidity for 24 hours, followed by an additional 24 hours at room temperature. Following incubation, the agar was removed from the plates and the motility diameters were measured in mm. Diameter measurements (S1 Data) were graphed and analyzed in GraphPad Prism.

### Live-imaging and tracking of *P. aeruginosa* directional response to *S. aureus*

To visualize single-cell motility behavior and measure cAMP in individual cells, *P. aeruginosa* cells were imaged using a previously described method [3,60]. *P. aeruginosa* and *S. aureus* were grown in M8T medium overnight at 37°C with aeration, subcultured the next day in fresh M8T, and grown to mid-log phase at 37°C with aeration. Cultures were standardized to $OD_{600}$ of 0.015 to 0.03 for *P. aeruginosa* or 0.05 to 0.1 for *S. aureus*. For coculture experiments, *P. aeruginosa* and *S. aureus* were mixed 1:1. One μL of mono- or coculture cells were inoculated onto a 10-mm diameter glass coverslip in a 35-mm dish before placing an agarose pad on top. Pads were made by pipetting 920 μL of M8T with 2% molten agarose into a 35-mm dish containing a 10-mm diameter mold and drying uncovered for 1 hour at room temperature,

followed by 1 hour 15 minutes at room temperature covered with a lid, then 1 hour at 37˚C before transferring the pad onto the inoculated coverslip. Time-lapse imaging was performed with an inverted Nikon Ti2 Eclipse microscope, 100× oil objective (1.45 NA), and Andor Sona camera. Phase contrast images were immediately acquired every 20 minutes for 2 hours, then every 1 second for 3 to 4 hours with 100 ms exposure and 20% DIA LED light. Fluorescent images were acquired every 20 minutes with TxRed images taken at 100 ms exposure and 20% Sola fluorescent light and YFP imaged at 25 ms exposure and 20% Sola fluorescent light.

Images were analyzed using Nikon NIS-Elements AR software. To automatically track movements of single cells, bacterial cells in the phase channel were first converted into binary objects by thresholding to the dark bacterial cells. The Tracking Module in NIS-Elements was then used to form trajectories for all binary objects. Specifically, the following parameters were used: Objects have a minimum area of 1 μm², track with random motion model, find center of object based on area, no maximum speed limit of each object track, allow new tracks after the first frame in file, and track objects backwards to previous frames. Tracks were allowed up to 60 gaps (frames) per track, and any tracks with less than 180 frames were automatically removed from the final trajectories. Trajectories were exported (S2–S7 Data) and analyzed in Python using the parameters described below and codes found in S1–S3 Codes.

## Principal direction of single-cell trajectories

The gyration tensor, commonly employed in polymer physics, determines the principal direction of each single-cell trajectory $i$ [61]. The shape of each random walk is described by the gyration tensor.

$$G_i = \frac{1}{N} \sum_{t=0}^{N} (\overrightarrow{r}(t) - \overrightarrow{r}_i^{cm})(\overrightarrow{r}(t) - \overrightarrow{r}_i^{cm}), \tag{1}$$

for a trajectory of $N$ time steps each at position $\overrightarrow{r}(t)$ at time $t$ and center of mass position $\overrightarrow{r}_i^{cm}$. Eigendecomposition of the gyration tensor determines the largest eigenvalue, and the associated eigenvector $\hat{e}_i^{\parallel}$ is identified as the principal direction of motion for the $i^{th}$ trajectory. The sign of $\hat{e}_i^{\parallel}$ is chosen by setting the direction of motion parallel to the end-to-end vector. The principal direction is determined without any information about the location of *S. aureus* colonies or other *P. aeruginosa* cells. Rose graphs shown in Figs 2–4 present histograms of $\hat{e}_i^{\parallel}$.

## Directed mean-squared displacement

The mean-squared displacement (MSD)

$$\Delta r^2(\Delta t) = \langle \Delta \overrightarrow{r}(\Delta t; t) \cdot \Delta \overrightarrow{r}(\Delta t; t) \rangle \tag{2}$$

of displacement vectors $\Delta \overrightarrow{r}(\Delta t; t) = \overrightarrow{r}(t + \Delta t) - \overrightarrow{r}(t)$ is a measure of the distance gone after a lag time $\Delta t$ (which we refer to as the "time step" in the main text), where the average $\langle \cdot \rangle$ is the ensemble average over all trajectories $i$ and all times $t$ within that trajectory. While the MSD quantifies the degree of motion, it assumes isotropic dynamics and so does not discern between any potential directionality in microbial motion. Quantifying the degree of motion parallel and perpendicular to the principal direction requires decomposing the movement into components parallel and perpendicular to each principal direction $\hat{e}_i^{\parallel}$. The displacement in the principal direction is $\Delta \overrightarrow{r}_{\parallel} = (\Delta \overrightarrow{r} \cdot \hat{e}_i^{\parallel}) \hat{e}_i^{\parallel}$ and the orthogonal direction is

$\Delta \overrightarrow{r}_{\perp} = \Delta \overrightarrow{r} - \Delta \overrightarrow{r}_{\parallel}$. From these, the parallel and perpendicular MSD are

$$\Delta r_{\parallel}^2(\Delta t) = \langle \Delta \overrightarrow{r}_{\parallel} \cdot \Delta \overrightarrow{r}_{\parallel} \rangle$$

$$\Delta r_{\perp}^2(\Delta t) = \langle \Delta \overrightarrow{r}_{\perp} \cdot \Delta \overrightarrow{r}_{\perp} \rangle. \tag{3}$$

To characterize the MSDs, the anomalous exponent $\alpha$ defined by the power law

$$\Delta r^2(\Delta t) \sim \Delta t^{\alpha}. \tag{4}$$

Exponents of $\alpha = 1$ describe diffusive dynamics, while $\alpha < 1$ represents subdiffusive motion and $\alpha > 1$ superdiffusive.

## Displacement probability distributions

The MSD can be deceiving since it is known that a broad class of diffusive dynamics exist in soft matter, biological, and complex systems for which the dynamics are "Brownian yet non-Gaussian" [62,63]. In such systems, the MSD appears diffusive with anomalous exponent $\alpha = 1$, but the probability density function (PDF) of steps $P(\Delta r; \Delta t)$ is non-Gaussian, a traditional assumption for Brownian motion. The probability of finding a bacterium $\Delta r$ after some lag time $\Delta t$ is called the van Hove self-correlation function, and it has proven useful in understanding the dynamics of simulations of twitching bacteria [24]. The formal definition of the van Hove self-correlation function is

$$P(\Delta \overrightarrow{r}; \Delta t) = \frac{1}{N} \langle \sum_{j=1}^{N} \delta(\Delta \overrightarrow{r} - [\overrightarrow{r}(t + \Delta t) - \overrightarrow{r}(t)]) \rangle. \tag{5}$$

To consider the probability that *P. aeruginosa* cells move a given distance towards *S. aureus* colonies, the van Hove function is found for steps parallel or orthogonal to the principal direction of motion, $\Delta \overrightarrow{r}_{\parallel}$ and $\Delta \overrightarrow{r}_{\perp}$, respectively. There are 2 probability density functions of particular interest here.

i. The first is a Gaussian diffusive distribution.

$$P_G(\Delta r; \Delta t) = \frac{1}{(4\pi D \Delta t)^{\frac{1}{2}}} exp\left(-\frac{\Delta r^2}{4D\Delta t}\right), \tag{6}$$

for a random process with diffusion coefficient $D$. This form can be scaled in time to collapse the distribution to $\tilde{P}_G(\Delta r) \sim e^{-\Delta \tilde{r}^2}$, where $\tilde{P}_G = \Delta t^{1/2} P_G$ and $\Delta \tilde{r} = \Delta t^{-1/2} \Delta r$. Thus, observing a Gaussian distribution at only one lag time is insufficient for determining Fickian diffusivity. An example of this is the microscopy imaging noise (S4A Fig): Although the step size distributions are Gaussian, they do not scale in time for lag times $\Delta t < 300s$, a distinct indication that this is a measure of the tracking noise and not diffusion of the dust particle.

ii. The second probability density functions of interest is a Laplace distribution.

$$P_L(\Delta r; \Delta t) = \frac{1}{2\lambda} exp\left(-\frac{|\Delta r|}{\lambda}\right), \tag{7}$$

for a decay length $\lambda$. Laplace distributions have longer tails than Gaussian distributions and have emerged as a canonical example of non-Gaussian functions that lead to Brownian MSDs [62,63]. If the decay length scales with lag time as $\lambda = (\langle D \rangle \Delta t)^{1/2}$ for an average

diffusivity $\langle D \rangle$, then the MSD scales diffusively. In this case, the distributions can be collapsed with the same scaling as the Gaussian distributions, $\tilde{P}_L(\Delta r) \sim e^{-|\Delta \tilde{r}|}$, where $\tilde{P}_L = \Delta t^{1/2} P_L$ and $\Delta \tilde{r} = \Delta t^{-1/2} \Delta r$.

The probability density distributions of step sizes are typically dominated by a sharp narrow peak of highly likely small step sizes, which represents jiggling and long tails of rare-but-large step sizes (Figs 2–4) [18]. The long tails are primarily exponential. It is tempting to think that the narrow peak of small steps sizes represents the imaging uncertainty. However, we find that the standard deviation of the imaging distribution is $1.4 \times 10^{-2} \mu m$ (S4A Fig), narrower than the width of the primary van Hove peak (S4B Fig). Indeed, the narrow peak is not Gaussian at all, but rather better fit by a second Laplace distribution. Thus, the displacement probability density distributions are well described as double Laplace functions (S4B Fig).

## Identifying subpopulations of persistent movers and resters

Qualitative assessment of the microscopy data makes it apparent that *P. aeruginosa* cells possess 2 dynamic modes:

1. Persistent "Resters": Colony-associate *P. aeruginosa* cells do not exhibit an active exploratory motion. Instead, the motion of these cells is composed of small "jiggling" and expansion due to colony growth.

2. Persistent "Movers": These are cells that have left the colony to actively move through the surroundings, either as individuals or in multicell rafts. Like "resters," these "movers" exhibit small jittering motion but also intermittently persistent motion. The intermittency of the motion can at times have a run-reversal-type or a run-rest-type character, but a "mover" is not simply a continually moving bacterium.

Both of these dynamic modes are composed of small jittery motion and larger motions, which makes it difficult to algorithmically separate the cells into subpopulations of movers and resters.

To disentangle the subpopulations, we consider the velocity–velocity correlation function $C_i^{vv}(\Delta t)$ for lag time $\Delta t$ for each bacterium $i$ averaged over start times. The velocity autocorrelation function takes into account both the direction and speed of the bacteria. Rather than averaging the correlation function over all possible start times within trajectory $i$, we employ a rolling velocity autocorrelation

$$C_i^{vv}(\Delta t; t) = |\langle \overrightarrow{v}(t + \tau) \cdot \overrightarrow{v}(t + \tau + \Delta t) \rangle_{\tau \leq T}|, \tag{8}$$

where $t$ is each time point in the trajectory, $T$ is the rolling window duration, $\tau$ is every possible starting time within the rolling window, and the average $\langle \cdot \rangle_{\tau \leq T}$ is over all starting times. The signed velocity autocorrelation function does not show anticorrelations, indicating that run-reversal dynamics are not statistically significant here. If the duration $T$ is too short, then the correlation functions are overly noisy, but if it is too long, then instances of correlated motion are smeared out. To assess the immediate degree of correlation in motion, the correlation function is averaged over the duration to produce a correlation constant $c_{vv}^T(t) = \langle C_i^{vv}(\Delta t; t) \rangle_{\Delta t \leq T}$. The correlation constant acts as a signal of immediately persistent motion, with persistent resters showing near-zero $c_{vv}^T(t)$ and movers having significantly larger values above a cutoff $c^*$. While $c_{vv}^T(t)$ matches our expectations from qualitative observations of the microscopy movies, false positive instances of colony-associated cells occur. Thus, the

signal is weighted by the behavior of neighboring cells

$$s(t; T, c^*, R) = \langle \Theta(c_{vv}^T(t) - c^*) \rangle_{r \leq R}, \tag{9}$$

where $\Theta(\cdot)$ is the Heaviside step function and the average is over all neighboring cells in the vicinity of $r \leq R$. Finally, the neighbor-weighted signal $s$ is given a cutoff $s^*$, above which cells are identified as "persistent movers" and below which they are "persistent resters." The parameters are chosen to be $T = 30s$, $c^* = 0.01$, $R = 6\mu m$, and $s^* = 0.75$ for this study. Due to the intermittent nature of the twitching dynamics, once a bacterium has been identified as a mover, it keeps a mover-designation until the trajectory is lost.

## Quantification of intracellular cAMP

To quantify cAMP in individual cells, time-lapse imaging was performed with *P. aeruginosa* cells carrying the P*xphA*-*yfp* P*rpoD*-*mKate2* dual fluorescent reporter [26]. Thresholding of bacterial cells in the red channel to the constitutively expressed P*rpoD*-*mKate2* fluorescence generated binary objects. The fluorescence of these binaries was then measured in the YFP channel for levels of P*xph*A-*yfp* expression. For total cAMP in a frame at a given time point, the ratio of YFP over mKate2 intensity for each bacterial cell was calculated, then summed with all other bacterial cells. For normalization, the total YFP/mKate2 ratio from all objects in the frame was normalized to the average area of all binary objects in the same frame. The ratios for each time point (S1 Data) were then graphed and analyzed in GraphPad Prism.

## Western blot of PilJ

*P. aeruginosa* strains with C-terminally His-tagged PilJ at the native site in the chromosome were incubated overnight in LB with aeration at 37°C, subcultured 1:100 in LB the following morning, then standardized to OD$_{600}$ 0.01 in LB before spread plating 100 μL on LA. Plates were incubated overnight at 37°C. The following day, cells were scraped up in 1 mL of phosphate buffered saline (PBS), pelleted and washed once in PBS, then resuspended in 1 mL PBS. The OD$_{600}$ for each cell suspension was measured, and an aliquot of 1 mL of cells at OD$_{600}$ 10 were pelleted on ice. Pellets were resuspended in RIPA buffer (50 mM Tris (pH 7.9), 150 mM NaCl, 1% NP-40, 0.5% Tween-20, 0.1% SDS) and briefly sonicated on ice to lyse cells. Proteins were standardized to 25 ug, following a bicinchoninic acid (BCA) assay, run on a 4% to 15% Tris-glycine polyacrylamide gel, then transferred onto a PVDF membrane. The membrane was blocked with 10% skim milk, and PilJ proteins were detected by chemiluminescence via incubation with primary mouse anti-His antibodies (Thermo Fisher), followed by incubation with goat-anti mouse, HRP (Thermo Fisher). The loading control was detected by incubation with StrepTactin-HRP conjugate (Bio-Rad). Blots were imaged using an Azure Sapphire.

## Supporting information

**S1 Fig. Complementing Δ*pilK*, and Δ*chpB* restores pilus-mediated chemotaxis.** Directional motility towards *S. aureus* secreted factors of wild type, Δ*pilK*, and Δ*chpB* with and without complementing plasmids carrying arabinose-inducible copies of *pilK* or *chpB*. Complemented strains were induced with 0.2% arabinose; however, phenotypes were the same in the absence of induction. Directional motility for at least 4 biological replicates, each containing a minimum of 3 technical replicates are shown. Statistical significance was determined with a one-way ANOVA followed by Dunnett's multiple comparisons test. *** indicates $p < 0.001$; ** indicates $p < 0.01$; *ns* indicates no statistically significant difference. The underlying data can be found in S1 Data. (TIFF)

**S2 Fig. Mean square displacements of wild type, Δ*chpB*, and Δ*pilK* trajectories.** MSDs for the parallel (‖) and perpendicular (⊥) directions of wild type, Δ*chpB*, and Δ*pilK* across different time steps (lag times, $\Delta t$). The anomalous diffusion exponent ($\alpha$) for each MSD is shown. Insets show the corresponding rose graphs from Fig 2B. The underlying data can be found in S2–S4 Data.
(TIFF)

**S3 Fig. van Hove distributions for all mutants.** Step size distributions for each *P. aeruginosa* strain are displayed by distance of displacement ($\Delta r$) of cells. Step size PDFs are shown for movers (solid lines) and resters (dotted lines) in the parallel (darker lines, ‖) and perpendicular (lighter lines, ⊥) directions. Step sizes for each *P. aeruginosa* strain were calculated from cell trajectories with a 1-second (left), 10-second (middle), and 100-second (right) time step ($\Delta t$). Right- and left-facing triangles (movers, ‖+ and ‖−, respectively) and up- and down-facing triangles (movers, ⊥+ and ⊥−, respectively) highlight the nonzero sharp-shoulder peak step size, when present. The underlying data can be found in S2–S6 Data.
(TIFF)

**S4 Fig. Dynamics due to imaging errors.** (**A**) MSDs for the parallel (‖) and perpendicular (⊥) directions for dust particles used to measure noise in the imaging. Inset shows the particle-displacement PDF. The PDF is a narrow noise peak that is fit to a Gaussian distribution (solid red line) but is nondiffusive, as it does not broaden in time. (**B**) PDF of the total cell step displacements ($\Delta r$), regardless of principal direction, for wild type cells at a time step of $\Delta t = 10$ seconds (black dots). The PDF is composed of a narrow peak of small displacements (jiggling) and long tails of large-but-rare displacements. The narrow peak cannot be explained by imaging uncertainty (solid red curve) and is better described by a Laplace distribution (Eq 7; solid yellow line), as are the long tails (solid dark blue line). The underlying data can be found in S1 and S7 Data.
(TIFF)

**S5 Fig. *P. aeruginosa* Δ*pilK* has bimodal pilus-mediated motility.** Separate Δ*pilK* cultures plated onto 4 individual experimental dishes and 4 fields of view in each culture dish were simultaneously imaged at 5 hours postinoculation. Agarose pads were made from the same media and dried under the same conditions at the same time. A range of motility phenotypes are seen between all microcolonies imaged.
(TIFF)

**S6 Fig. PilJ protein levels are stable in all mutants.** Western blot of His-tagged WT or mutant PilJ protein derived from *P. aeruginosa*. An untagged protein sample from *P. aeruginosa* WT was used as a control. Full-length PilJ is expected to be 75 kDa, while the PilJ protein lacking the LBDs is expected to be 46 kDa. Detection was performed for PilJ::6xHis alone (**A**) and for PilJ::6xHis with a loading control (**B**).
(TIF)

**S7 Fig. Methylation sites of PilJ.** Amino acid sequence of *P. aeruginosa* PA14 PilJ (PA14_05360) with conserved MCP methylation motifs highlighted in pale orange and predicted methyl modification glutamate/glutamine residue pairs in bold.
(TIFF)

**S8 Fig. Complemented *pilJ*<sub>Q412A, E413A</sub> with full-length PilJ displays bipolar PilJ localization and restoration of TFP-mediated directional motility.** (**A**) Representative *P. aeruginosa* cells with bipolarly localized GFP-tagged PilJ. (**B**) Directional motility towards *S. aureus* secreted factors of wild type or *pilJ*<sub>Q412A, E413A</sub> with and without complementing plasmids

carrying arabinose-inducible copy of wild type *pilJ*. Complemented strains were induced with 0.2% arabinose; however, phenotypes were the same in the absence of induction. Directional motility for at least 3 biological replicates, each containing a minimum of 3 technical replicates are shown, and statistical significance was determined with a one-way ANOVA followed by Dunnett's multiple comparisons test to compare each strain to wild type *P. aeruginosa* carrying p-araBAD-*pilJ*. **** indicates $p < 0.0001$; *ns* indicates no statistically significant difference. The underlying data can be found in S1 Data.
(TIFF)

**S9 Fig. Complemented *pilJ*$_{\Delta LBD1-2}$ has bipolarly localized PilJ but only partial restoration of TFP-mediated chemotaxis.** (**A**) Representative *P. aeruginosa* cells with bipolarly localized GFP-tagged PilJ. (**B**) Directional motility towards *S. aureus* secreted factors of wild type or *pilJ*$_{\Delta LBD1-2}$ with and without complementing plasmid carrying arabinose-inducible copy of wild type *pilJ*. Complemented strains were induced with 0.2% arabinose; however, phenotypes were the same in the absence of induction. Directional motility for at least 3 biological replicates, each containing a minimum of 3 technical replicates are shown, and statistical significance was determined with a one-way ANOVA followed by Dunnett's multiple comparisons test to compare each strain to wild type *P. aeruginosa* carrying p-araBAD-*pilJ*. **** indicates $p < 0.0001$; ** indicates $p < 0.01$; *ns* indicates no statistically significant difference. The underlying data can be found in S1 Data.
(TIFF)

**S10 Fig. PilJ mutant lacking a portion of the periplasmic region is nonmotile.** Twitching motility diameters of *P. aeruginosa* wild type, Δ*pilJ*, and a *pilJ* mutant lacking amino acids 80-273 (*pilJ*$_{\Delta 80-273}$). Macroscopic motility measurements are shown for 2 biological replicates, each containing 4 technical replicates, and statistical significance was determined with a one-way ANOVA followed by Dunnett's multiple comparisons test. **** indicates $p < 0.0001$. The underlying data can be found in S1 Data.
(TIFF)

**S11 Fig. Enzymes that control cAMP levels are necessary for pilus-mediated chemotaxis.** Directional motility towards *S. aureus* secreted factors of Δ*cyaB* and Δ*cpdA*. At least 3 biological replicates, each containing a minimum of 3 technical replicates are shown, and statistical significance was determined with a one-way ANOVA followed by Dunnett's multiple comparisons test. **** indicates $p < 0.0001$. The underlying data can be found in S1 Data.
(TIFF)

**S1 Video. *P. aeruginosa* wild type in monoculture.** Duration 3.5 hours. Three hours post-inoculation. Acquisition interval 1 second. Output interval every 40th frame at 50 ms/frame.
(MOV)

**S2 Video. *P. aeruginosa* wild type in coculture.** Duration 1 hour 45 minutes. Three hours post-inoculation. Acquisition interval 1 second. Output interval every 40th frame at 50 ms/frame.
(MOV)

**S3 Video. *P. aeruginosa* Δ*chpB* in monoculture.** Duration 2 hours. Three hours post-inoculation. Acquisition interval 1 second. Output interval every 40th frame at 50 ms/frame.
(MOV)

**S4 Video. *P. aeruginosa* Δ*chpB* in coculture with *S. aureus* wild type.** Duration 3 hours. Two hours postinoculation. Acquisition interval 1 second. Output interval every 40th frame at 50 ms/frame.
(MOV)

**S5 Video. *P. aeruginosa* Δ*pilK* in monoculture.** Growing microcolony cells. Duration 3 hours. Three hours postinoculation. Acquisition interval 1 second. Output interval every 40th frame at 50 ms/frame.
(MOV)

**S6 Video. *P. aeruginosa* Δ*pilK* in monoculture.** Hypermotile cells. Duration 3 hours. Three hours postinoculation. Acquisition interval 1 second. Output interval every 40th frame at 50 ms/frame.
(MOV)

**S7 Video. *P. aeruginosa* Δ*pilK* in coculture with *S. aureus* wild type.** Growing microcolony cells. Duration 4 hours. Two hours postinoculation. Acquisition interval 1 second. Output interval every 40th frame at 50 ms/frame.
(MOV)

**S8 Video. *P. aeruginosa* Δ*pilK* in coculture with *S. aureus* wild type.** Hypermotile cells. Duration 4 hours. Two hours postinoculation. Acquisition interval 1 second. Output interval every 40th frame at 50 ms/frame.
(MOV)

**S9 Video. *P. aeruginosa* Δ*pilJ* Δ*flgK* in coculture with *S. aureus* wild type.** Duration 1 hour 15 minutes. Three hours postinoculation. Acquisition interval 1 second. Output interval every 40th frame at 50 ms/frame.
(MOV)

**S10 Video. *P. aeruginosa* Δ*chpB* Δ*flgK* in coculture with *S. aureus* wild type.** Duration 3 hours. Three hours postinoculation. Acquisition interval 1 second. Output interval every 40th frame at 50 ms/frame.
(MOV)

**S11 Video. *P. aeruginosa* Δ*pilK* Δ*flgK* in coculture with *S. aureus* wild type.** Growing microcolony cells. Duration 2.5 hours. Three hours postinoculation. Acquisition interval 2 seconds. Output interval every 20th frame at 50 ms/frame.
(MOV)

**S12 Video. *P. aeruginosa* Δ*pilK* Δ*flgK* in coculture with *S. aureus* wild type.** Hypermotile cells. Duration 2.5 hours. Three hours postinoculation. Acquisition interval 2 seconds. Output interval every 20th frame at 50 ms/frame.
(MOV)

**S13 Video. *P. aeruginosa* pilJ$_{Q412A, E413A}$ in monoculture.** Duration 2 hours 40 minutes. Two hours 20 minutes postinoculation. Acquisition interval 1 second. Output interval every 40th frame at 50 ms/frame.
(MOV)

**S14 Video. *P. aeruginosa* pilJ$_{Q412A, E413A}$ in coculture with *S. aureus* wild type.** Duration 3 hours. Two hours postinoculation. Acquisition interval 1 second. Output interval every 40th frame at 50 ms/frame.
(MOV)

**S15 Video. *P. aeruginosa* pilJ$_{Q412A, E413A}$ Δ*flgK* in coculture with *S. aureus* wild type.** Duration 3.5 hours. One hour postinoculation. Acquisition interval 1 second. Output interval every 40th frame at 50 ms/frame.
(MOV)

**S16 Video.** *P. aeruginosa pilJ*$_{\Delta LBD1-2}$ **in monoculture.** Duration 2 hours 40 minutes. Two hours 20 minutes postinoculation. Acquisition interval 1 second. Output interval every 40th frame at 50 ms/frame.
(MOV)

**S17 Video.** *P. aeruginosa pilJ*$_{\Delta LBD1-2}$ **in coculture with** *S. aureus* **wild type.** Duration 2 hours. Three hours postinoculation. Acquisition interval 1 second. Output interval every 40th frame at 50 ms/frame.
(MOV)

**S18 Video.** *P. aeruginosa pilJ*$_{\Delta LBD1-2}$ *ΔflgK* **in coculture with** *S. aureus* **wild type.** Duration 3 hours. Two hours postinoculation. Acquisition interval 1 second. Output interval every 40th frame at 50 ms/frame.
(MOV)

**S1 Table. Strains and key resources used in this study.**
(XLSX)

**S2 Table. Oligonucleotides used in this study.**
(XLSX)

**S1 Data. Replicates for macroscopic motility and cAMP quantification.**
(XLSX)

**S2 Data. Trajectories for** *P. aeruginosa* **wild type.**
(XLSX)

**S3 Data. Trajectories for** *P. aeruginosa* *ΔchpB*.
(XLSX)

**S4 Data. Trajectories for** *P. aeruginosa* *ΔpilK*.
(XLSX)

**S5 Data. Trajectories for** *P. aeruginosa pilJ*$_{Q412A, E413A}$.
(XLSX)

**S6 Data. Trajectories for** *P. aeruginosa pilJ*$_{\Delta LBD1-2}$.
(XLSX)

**S7 Data. Trajectories for imaging error analysis.**
(XLSX)

**S1 Code. Random walk analysis functions.**
(PY)

**S2 Code. Separate movers and resters analysis.**
(PY)

**S3 Code. Graph format.**
(PY)

**S1 Raw Image. Original western blot image for S6 Fig.**
(PDF)

## Acknowledgments

We thank Drs. George O'Toole and Sherry Kuchma for *P. aeruginosa* and *E. coli* bacterial strains. We also thank Dr. J. Muse Davis for use of the stereoscope and Dr. Timothy Yahr for the pEXG2-Tc cloning vector. We are grateful to George O'Toole and members of the Limoli Lab for thoughtful discussions and feedback on the manuscript.

## Author Contributions

**Conceptualization:** Kaitlin D. Yarrington, Tyler N. Shendruk, Dominique H. Limoli.

**Data curation:** Tyler N. Shendruk.

**Formal analysis:** Kaitlin D. Yarrington, Tyler N. Shendruk.

**Funding acquisition:** Kaitlin D. Yarrington, Tyler N. Shendruk, Dominique H. Limoli.

**Investigation:** Kaitlin D. Yarrington.

**Methodology:** Kaitlin D. Yarrington, Tyler N. Shendruk, Dominique H. Limoli.

**Resources:** Tyler N. Shendruk, Dominique H. Limoli.

**Software:** Tyler N. Shendruk.

**Supervision:** Dominique H. Limoli.

**Visualization:** Kaitlin D. Yarrington.

**Writing – original draft:** Kaitlin D. Yarrington, Tyler N. Shendruk, Dominique H. Limoli.

**Writing – review & editing:** Kaitlin D. Yarrington, Tyler N. Shendruk, Dominique H. Limoli.

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
