## [Editor Report · Decision Letter 0]

14 Nov 2023

Dear Dr. Limoli, 

Thank you for submitting your manuscript entitled "The type IV pilus chemoreceptor PilJ controls interspecies bacterial chemotaxis" for consideration as a Research Article by PLOS Biology.

Your manuscript, previous reviews and your responses to reviewers have now been evaluated by the PLOS Biology editorial staff, as well as by an academic editor with relevant expertise, and I am writing to let you know that we would like to send consider your manuscript further without the need for further peer review. 

However, before we can consider your manuscript further, we need you to complete your submission by providing the metadata that is required for full assessment. To this end, please login to Editorial Manager where you will find the paper in the 'Submissions Needing Revisions' folder on your homepage. Please click 'Revise Submission' from the Action Links and complete all additional questions in the submission questionnaire.

Once your full submission is complete, your paper will undergo a series of checks. After your manuscript has passed the checks we will send you a decision letter. To provide the metadata for your submission, please Login to Editorial Manager (https://www.editorialmanager.com/pbiology) within two working days, i.e. by Nov 16 2023 11:59PM.

Kind regards,

Paula

---

Senior Editor

PLOS Biology

---

## [Editor Report · Decision Letter 1]

17 Nov 2023

Dear Dr. Limoli,

***EITHER***

Thank you for your patience while your manuscript "The type IV pilus chemoreceptor PilJ controls interspecies bacterial chemotaxis" was being assessed at PLOS Biology. Your manuscript, previous reviews and your responses to reviewers have now been evaluated by the PLOS Biology editors and an Academic Editor with relevant expertise.

In light of our Academic Editor assessment, that you can find at the end of this letter, we are pleased to offer you the opportunity to address the comments from the Academic Editor in a revision that we anticipate should not take you very long. In particular, we would be interested in considering a revised version of the manuscript in which you discuss, compare and contrast the behavioural mechanism that results in chemotaxis in P. aeruginosa towards PSMs to the regulation of flagella-based motility.

Please also address the following policy and editorial requests:

1. DATA POLICY:

A) Supplementary files (e.g., excel). Please ensure that all data files are uploaded as 'Supporting Information' and are invariably referred to (in the manuscript, figure legends, and the Description field when uploading your files) using the following format verbatim: S1 Data, S2 Data, etc. Multiple panels of a single or even several figures can be included as multiple sheets in one excel file that is saved using exactly the following convention: S1_Data.xlsx (using an underscore).

B) Deposition in a publicly available repository. Please also provide the accession code or a reviewer link so that we may view your data before publication. 

Regardless of the method selected, please ensure that you provide the individual numerical values that underlie the summary data displayed in the following figure panels as they are essential for readers to assess your analysis and to reproduce it: We are aware that you already provided some of the data, but we require this for: Figures 2BCD, 3BCD, 4BCDE, 5DEF, and Supplementary Figures F2-SF1, F2-SF2, F2-SF3, F2-SF4AB, F3-SF2B, F4-SF1B, F4-SF2, F5-SF1.

2. CODE POLICY

Per journal policy, as the code that you have generated is important to support the conclusions of your manuscript, we require that you make it available without restrictions upon publication. Please ensure that the code is sufficiently well documented and reusable, and that your Data Statement in the Editorial Manager submission system accurately describes where your code can be found.

3. We suggest a change in the title to make it accessible to a broader audience: "The type IV pilus chemoreceptor PilJ controls chemotaxis of one bacterial species towards another"

**IMPORTANT - SUBMITTING YOUR REVISION**

*Resubmission Checklist*

*Published Peer Review*

*PLOS Data Policy*

*Blot and Gel Data Policy*

Sincerely,

Paula

---

Senior Editor

PLOS Biology

EDITED COMMENTS FROM THE ACADEMIC EDITOR:

Based on the reviews of reviewers #1 and #3 and the authors' responses, I am very much in favour of this manuscript for PLOS Biology. The authors also respond very convincingly to the more critical comments of reviewer #2.

The elucidation of the behavioural mechanism that results in chemotaxis towards PSMs is very interesting, novel and of broad interest, however, the discussion of this finding is not very elaborate.

The authors should discuss, compare and contrast the behavioural mechanism that results in chemotaxis in P. aeruginosa towards PSMs to the regulation of flagella-based motility.

---

## [Editor Report · Decision Letter 2]

5 Jan 2024

Dear Dr Limoli,

Thank you for the submission of your revised Research Article "The type IV pilus chemoreceptor PilJ controls chemotaxis of one bacterial species towards another" for publication in PLOS Biology. I have taken over handling your manuscript as my colleague Dr Paula Jauregui has sadly left PLOS Biology. On behalf of my colleagues and the Academic Editor, Lotte Søgaard-Andersen, I'm pleased to say that we can in principle accept your manuscript for publication, provided you address any remaining formatting and reporting issues. These will be detailed in an email you should receive within 2-3 business days from our colleagues in the journal operations team; no action is required from you until then. Please note that we will not be able to formally accept your manuscript and schedule it for publication until you have completed any requested changes.

IMPORTANT: Please note that after discussion with the team, I've changed the title of your article to "The type IV pilus chemoreceptor PilJ controls chemotaxis of one bacterial species towards another" (the short title is unchanged). We felt that this would make the findings more accessible to the wider readership. Do let me know if this is problematical.

Sincerely, 

Roli Roberts

Senior Editor

PLOS Biology

rroberts@plos.org